# The crucial but insufficient role of E2s domain's residues 490 and 492 in determining the host tropism of hepatitis E virus

Zi-Min Tang[1,2,3,6], Cheng-Yu Yang[1,2,6], Gui-Ping Wen [4,6], Chang Liu[1,2,6], Dong Ying[1,2,6], Yang Huang[1,2], Zi-Hao Yu[1,2], Ming-Yu Li[1,2], Si-Ling Wang[1,2], Zi-Hao Chen [1,2], Jun-Fei Liu[1,2], Mu-Jin Fang[1,2,3], Ying-Bin Wang[1,2], Jun Zhang [1,2], Ying Gu [1,2], Hai Yu [1,2], Shao-Wei Li [1,2] ✉, Qing-Bing Zheng [1,2] ✉, Ning-Shao Xia [1,2,3] ✉ & Zi-Zheng Zheng [1,2,5] ✉

Hepatitis E virus (HEV) is a significant pathogen causing acute viral hepatitis globally, posing a particular threat to pregnant women. HEV infects a range of host species, with distinct genotypes exhibiting genotype-specific tropism for different host hepatocytes. The P domain of viral capsid protein plays a central role in host cell attachment, but the molecular determinants that govern its host specificity remain unclear. This study investigates the molecular mechanisms underlying HEV host tropism by using a zoonotic HEV specific antibody 6H8. An epitope involving residues 490 and 492 is identified crucial for both mAb 6H8 binding and virus-cell attachment. Structure-based muta-genesis, molecular dynamics simulations, virus-cell attachment assays, and viral infectivity assays highlight the importance of the N490 and M492 residues in maintaining the structural integrity of the 6H8 epitope, influencing host specificity. Mutations at 490 and 492 permit HEV-1's and disrupt HEV-4's binding and infection in porcine hepatocytes. However, they are insufficient alone for reestablishing swine infection in vivo, indicating additional factors are involved in HEV's host tropism. Our findings suggest that N490 and M492 are critical but not sole determinants of HEV-4's specific tropism for porcine hepatocytes and advance our understanding of HEV's zoonotic transmission and host specificity.

Hepatitis E virus (HEV), a positive-sense RNA virus, is a significant pathogen causing acute viral hepatitis globally. This virus poses a grave threat to pregnant women, with mortality rates reaching up to 25%[1]. HEV strains isolated from humans are categorized within the genus Orthohepevirus A, family Hepeviridae[2,3]. Phylogenetic studies have identified eight HEV genotypes, with genotypes 1 to 4 being prevalent globally, while genotypes 5 to 8 are relatively rare[3]. The first four genotypes are further grouped based on their primary hosts: geno-types 1 and 2 (HEV-1 and HEV-2) are exclusive to humans and non-human primates, forming the human group, while genotypes 3 and 4 (HEV-3 and HEV-4), comprising the zoonotic group, exhibit a broader host range that includes humans, swine, and other mammals[4].

Since the initial discovery of HEV in swine in 1997 in the United States[5], zoonotic strains have been detected in regions heavily reliant on pork, establishing swine as significant reservoirs of HEV[6,7]. HEV infections in swine have been reported in both developed and developing countries[8,9], with notably high prevalence rates of anti-HEV antibodies and HEV RNA reaching up to 92.8%[8] and 44.4%[9], respectively. The transmission of zoonotic HEV to humans, particularly from HEV-3 and HEV-4 through pork products, has been extensively documented[10–13]. The distinct host tropism between human and zoonotic HEV groups has led to different epidemiological patterns: human group HEVs typically cause periodic outbreaks in developing countries through contaminated drinking water[14], whereas zoonotic group HEVs are associated with sporadic cases in both developing and developed countries through meat consumption[15,16]. There is also the potential for occupational transmission among individuals in close contact with infected animals[17]. These differences suggest the existence of varying molecular mechanisms of host cell recognition, a topic that remains not fully understood.

The underlying viral and host factors responsible for the host range of HEV remain elusive[17]. Various viral factors, including open reading frame 1 (ORF1), ORF2, 5' noncoding region (NCR), codon usage, and adaptive evolution, have been reported to influence host range determination[18–27]. HEV virus-like particles (VLPs) bound and entered cells in a species-specific manner, suggesting that viral entry is a vital determinant of HEV host tropism and zoonotic potential[26]. The outer domain of pORF2, referred to as the P domain or E2s domain, facilitates the virion's attachment to host cells[25]. This suggests that the interaction of pORF2's with host cells plays a crucial role in determining the virus's host specificity. Previous study has emphasized the importance of pORF2, particularly the E2s domain, in enabling the virus to infect porcine kidney cells[24]. The chimeric virus with the E2s domain from HEV-1 integrated into the HEV-3 genome exhibited similar ability to infect swine kidney cells to that of HEV-1 and different from that of HEV-3[24]. These observations underscore the central role of the E2s in determining HEV's host tropism. Efforts to discern the differences in the E2s domain between human and zoonotic groups HEVs have identified amino acids with distinct patterns across the two groups, such as amino acids 497, 517, 527, 537, 554, 569, 571, 593, and 599 in pORF2[24]. However, none of these distinguished residues have been conclusively linked to the tropism of zoonotic HEV for swine[24]. Whether alteration in the E2s domain could disrupt the host's tropism of zoonotic HEV for swine has not yet been verified in vivo. Furthermore, the critical residues of the E2s domain responsible for the swine tropism of zoonotic HEV remain unknown.

Given the critical role of the E2s domain in determining HEV's host tropism, in this study we aimed to pinpoint specific epitope that differ between human and zoonotic groups of HEV. Utilizing a zoonotic HEV-specific monoclonal antibody (mAb) 6H8 and analyzing structures of various E2s and 6H8 immune complexes, we identified an epitope located exclusively on the zoonotic HEV viral capsid. Through a combination of structure-based mutagenesis, molecular dynamics simulations, virus-cell attachment assays and viral infectivity assays, we identify key residues (S488, T489, N490, and Y532) in the E2 protein that mediate both antibody recognition and virus-cell attachment. We further show how mutations in these residues affect HEV's ability to bind specifically to porcine hepatocytes, revealing important insights into the viral determinants of host specificity. Structural and molecular dynamics analyses reveal that the N490 residue plays a pivotal role in stabilizing the epitope's conformation, maintaining the structural integrity of the 480- and 530-loops. Mutations at N490 or M492 disrupt the 6H8 epitope, significantly reducing binding to porcine hepatocytes and impairing viral tropism. These insights offer a deeper understanding of the viral-host interactions that underlie HEV infection.

## Results

### HEV-4 capsid specifically binds to porcine hepatocytes

Given that swine are the primary natural reservoir for zoonotic group HEV while no porcine hepatogenic cell lines are current available, primary porcine hepatocytes (PPH) were isolated and utilized as a model for HEV study, aligning with previous studies[28,29]. Due to the limited prevalence of HEV-2, we focused on native HEV-1, HEV-3 and HEV-4 viruses. Our study compared PPH with the human hepatogenic cell line HepG2 and primary human hepatocytes (PHH) to evaluate the binding and entry capabilities of HEV-1 (human group), HEV-3 (zoonotic group) and HEV-4 (zoonotic group) strains. The binding potency of the HEV-3 and HEV-4 virus to PPH was significantly higher than that of HEV-1, exceeding by more than 15-fold ($p < 0.01$) (Fig. 1A and Fig. S1). In contrast, the binding potency of the three viruses on HepG2 cells and PHH showed no notable differences (Fig. 1B, Fig. S1, and Fig. S2). The infectivity of native viruses in PPH and HepG2 were also analyzed using immunofluorescence assay (IFA). The results showed that infectivity of HEV-3 and HEV-4 was observed in both HepG2 cells and PPH, while infectivity of HEV-1 was observed exclusively in HepG2 cells but not in PPH (Fig. 1C, Fig. S3, and Fig. S4). All viruses in both HepG2 and PPH were neutralized by mAb A286 (Fig. S3, and Fig. S4), which recognizes all four HEV genotypes equally[30]. We then investigated the binding and entry potential of the widely reported HEV p239 VLPs on PPH, HepG2 and PHH, respectively, using a previously established IFA[25]. Consistent with the findings of native viruses, both genotype 1 p239 (p239-1) and genotype 4 p239 (p239-4) demonstrated effective binding and entry into HepG2 cell and PHH within 1 h and 4 h, respectively (Fig. 1D and Fig. S5). In contrast, only p239-4 was found to effectively bind and enter PPH cells (Fig. 1E). The binding and entry of p239-1 were observed exclusively in HepG2 and PHH but not in PPH cells (Fig. 1D, E, and Fig. S5). The binding and entry of VLP on HepG2, PPH, and PHH could be blocked by the mAb 8G12 (Fig. S5 and Fig. S6), which recognize all four HEV genotypes equally[31]. These findings suggests that HEV-1 and HEV-4 exhibit similar binding and entry capabilities in human hepatocytes, yet differ markedly in their interactions with porcine hepatocytes.

The observed difference in cell tropism suggested distinct binding and entry mechanisms for HEV-1 and HEV-4. In a previous study, we demonstrated that p239-1 consistently shares binding sites with the native HEV-1 virus on HepG2 cells[25]. Employing a VLP-virion blocking assay akin to our previous methods, we demonstrated that both p239-1 and p239-4 exhibit a comparable ability to inhibit the binding of HEV-1 to HepG2 cells, achieving half-maximal inhibitory concentrations ($IC_{50}$) of 21.13 μg/mL and 14.95 μg/mL, respectively (Fig. 1F). In contrast, p239-1 exhibited a significantly reduced capability, nearly tenfold less, in preventing HEV-4 from binding to HepG2 cells compared with p239-4 (Fig. 1G). Remarkably, p239-4 could completely prevent HEV-4's attachment to HepG2 cells with an $IC_{50}$ value of 8.88 μg/mL, whereas p239-1 could only partially (60%) block the binding of HEV-4 on HepG2 cells even at a high input concentration, resulting in an $IC_{50}$ of 94.69 μg/mL. These findings suggest that p239-4 competes more effectively with HEV-4 and HEV-1 for their binding sites on HepG2 cells than p239-1, the latter cannot fully occupy HEV-4's binding sites.

As for PPH, p239-4 demonstrated a moderate ability to block HEV-4 binding to PPH, with an $IC_{50}$ of 29.28 μg/mL, while no detectable blocking ability was detected for that of p239-1 (Fig. 1H). Consequently, we hypothesize that both porcine and hominine hepatocytes contain HEV-3 and HEV-4-specific binding sites, which can be recognized by HEV-3, HEV-4, and p239-4 through unique viral determinants that are not accessible to p239-1 or HEV-1.

### Characterization of a zoonotic HEV-specific mAb 6H8

To determine the specific determinant present on HEV-4, a zoonotic group HEV-specific mAb, 6H8, from a previously reported panel of mAbs against the E2s domain was selected for further analysis[32].

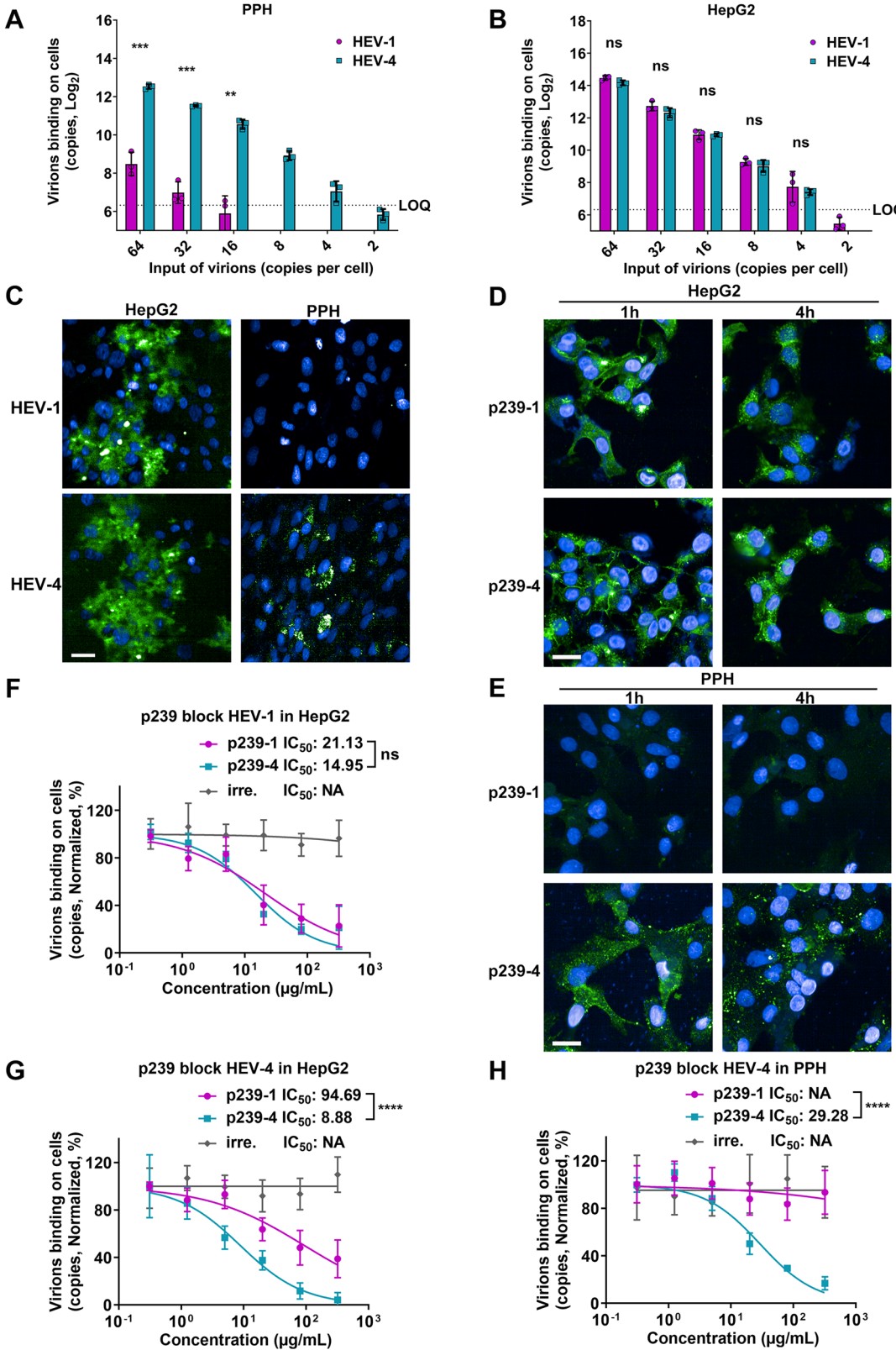

Binding assays and Western blot analyses indicated that mAb 6H8 specifically recognized the dimer (E2) and VLP (p239) forms of capsid proteins belonging to HEV-3 and HEV-4, but not to HEV-1 and HEV-2 (Fig. 2A, B and Fig. S7). Surface plasmon resonance (SPR) assays conformed that mAb 6H8 efficiently bound to the E2 domains of HEV-3 (E2-3) and HEV-4 (E2-4) with high affinities of 5.39 nM and 2.51 nM, respectively, while its binding affinities for E2-1 and E2-2 were too weak

to reach the detection limit (Fig. S8). These findings were corroborated by analytical ultracentrifugation (AUC) analysis, which detected immune-complexes only in the mixture of 6H8 with E2-3 and E2-4 (~10.06 S), but not with E2-1 and E2-2 (Fig. 2C-F).

We next evaluated the capture abilities of mAb 6H8 against native viruses using an immune capture assay as previously described[32], with the mAb 8G12[31] as a positive control. As expected, 6H8 demonstrated

**Fig. 1 | Binding potency of genotypes 1 and 4 HEV with primary porcine hepatocytes (PPH). A** Binding ability of genotype 1 HEV (HEV-1) and genotype 4 HEV (HEV-4) to PPH. Magenta column represents HEV-1, and cyan column represents HEV-4. *P* values were 0.0004, 0.0002, and 0.0011 for virion inputs of 64, 32, and 16 copies per cell, respectively. **$p < 0.01$; ***$p < 0.001$. Data represent three biological replicates. LOQ: limit of quantitation. **B** Binding ability of HEV-1 and HEV-4 to HepG2 cells. Magenta column represents HEV-1, and cyan column represents HEV-4. ns: not significant. Data represent three biological replicates. LOQ: limit of quantitation. **C** Infectivity of HEV-1 and HEV-4 in HepG2 and PPH detected by immunofluorescence assay. Green fluorescence indicates HEV ORF2 protein expression, and blue fluorescence indicates 4′,6-diamidino-2-phenylindole (DAPI) nuclear staining. Scale bar, 25 μm. **D** Binding (1 h) and penetration (4 h) of genotype 1 p239 (p239-1) and genotype 4 p239 (p239-4) to HepG2 cells detected by immunofluorescence analysis. Green indicates positive for p239 protein and blue indicates positive for DAPI staining. Scale bar, 25 μm. **E** Binding (1 h) and penetration (4 h) of p239-1 and p239-4 to PPH detected by immunofluorescence analysis. Green indicates positive for p239 protein and blue indicates DAPI staining. Scale bar, 25 μm. **F** HEV-1 binding on HepG2 cells blocked by different genotypes p239. Magenta lines represent HEV binding blocked by p239-1, cyan lines represent HEV binding blocked by p239-4, and gray lines represent HEV binding in the presence of an irrelevant protein (irre.). ns: not significant (p = 0.8108). NA, not analyzed. Data represent three biological replicates. **G** HEV-4 binding on HepG2 cells blocked by different genotypes p239. Magenta lines represent HEV binding blocked by p239-1, cyan lines represent HEV binding blocked by p239-4, and gray lines represent HEV binding in the presence of an irrelevant protein (irre.). ****$p < 0.0001$. NA, not analyzed. Data represent three biological replicates. **H** HEV-4 binding on PPH blocked by different genotypes p239. Magenta lines represent HEV binding blocked by p239-1, cyan lines represent HEV binding blocked by p239-4, and gray lines represent HEV binding in the presence of an irrelevant protein (irre.). ****$p < 0.0001$. NA, not analyzed. Data represent three biological replicates. Data are presented as mean ± SD (**A**, **B**, and **F**–**H**). Statistical analysis was performed with unpaired t-test (**A**, **B**) or two-way ANOVA (**F**–**H**), and the statistical test used was two-sided. Experiments in Fig. 1C–E were performed once. Source data are provided as a Source Data file.

capture abilities for HEV-3 and HEV-4 but not HEV-1, while 8G12 exhibited similar capture abilities for all three examined HEV genotypes (Fig. 2G, H). Furthermore, in vitro neutralization tests with HepG2 cells revealed that 6H8 specifically neutralized HEV-3 and HEV-4, with comparable $IC_{50}$ values of 19.18 μg/ml and 13.88 μg/mL in HepG2 (Fig. 2I) respectively. Conversely, 6H8 showed no significant neutralizing activity against HEV-1 in HepG2 ($IC_{50} > 2000$ μg/mL) (Fig. 2I). The highest 6H8 concentration even resulted in less than 50% neutralization (Fig. 2I). 6H8 exhibited comparable neutralizing activity against HEV-3 and HEV-4 in PPH, with $IC_{50}$ values of 22.03 μg/mL and 25.61 μg/mL, respectively, similar to those observed in HepG2 cells (Fig. 2J). In comparison, 8G12 displayed a consistent neutralization potency across the different HEV genotypes, with $IC_{50}$ values ranging from 8.92 μg/mL to 10.68 μg/mL in HepG2 and 13.78 μg/mL to 18.30 μg/mL in PPH (Fig. 2K, L). IFA confirmed that mAb 6H8 effectively neutralized HEV-3 and HEV-4 in both PPH and HepG2 cells, but failed to neutralize HEV-1 in HepG2 cells (Fig. S3 and Fig. S4). Collectively, these outcomes validate mAb 6H8 as a neutralizing antibody specific to the zoonotic group HEV.

**Crystal and cryo-EM structures of 6H8 immune complexes**

We subsequently sought to determine the epitope recognized by mAb 6H8 at atomic resolution. The crystal structure of the immune-complex of E2s-4 bound with the antigen-binding fragment (Fab) of 6H8 (E2s-4:6H8) was resolved at a resolution of 2.3 Å and refined to an R-factor of 21.6 % ($R_{free}$ = 23.6%) (Fig. 3A, Fig. S9, and Table S1). To verified the binding mode of 6H8 on HEV virus, we also prepared the p495 (amino acids 112-606) VLP of HEV-4 (p495-4) and complexed it with 6H8 Fab. The resultant immune complex (p495-4:6H8) was subjected to cryo-electron microscopy (cryo-EM) single-particle analysis, achieving a 4.1 Å density map (Fig. 3B, Fig. S10, and Table S2). In the asymmetric unit of the crystal structure, one dimer of E2s-4 was bound by two 6H8 Fab molecules (Fig. 3A), a configuration echoed in the cryo-EM structure, where the crystal structure fitting well within the cryo-EM density map (Fig. 3B). The 6H8 Fab exhibited a well-defined electron density map, closely resembling its crystal structure (Fig. S9). The Fab's interaction footprint, located on the dimer's outermost side, buries a total of ~650 Å$^2$ of surface area (Fig. 3C). The interaction between 6H8 and E2s-4 involved amino acids Y480, Q482, T483, S487-N490, I529-S533, Y559, N560, Y584, and Y585 of E2s-4 (Fig. 3C). The binding region of 6H8 comprised of all six complementarity determining region (CDR) loops (Fig. 3D), with key interactions facilitated by 13 hydrogen-bonding contacts (≤3.2 Å) among the main-chain and side-chain atoms (Table S3), alongside several van der Waals interactions.

Sequence analysis revealed that among the 16 epitope residues involved in 6H8 recognition, two residues (amino acids 483 and 490)

differentiate between the human (S483 and G490) and zoonotic (T483 and N490) groups of HEVs (Fig. 3E and Fig. S11), suggesting these residues are crucial for 6H8's specificity towards HEV-3 and HEV-4. The remaining epitope residues are conserved across all HEV genotypes. Additionally, our previous research identified two other neutralizing epitopes on E2s structurally[31,33]. Notably, the HEV-1-specific neutralizing epitope, recognized by mAb 8C11, is located in E2s-1 dimer's groove region (Fig. 3F), with a distinctive amino acid at position 497 crucial for its specificity. Another neutralizing epitope, recognized by mAb 8G12, was mapped to the E2s dimerization region and conserved across all HEV genotypes. The epitope for mAb 6H8, distinct from those of 8C11 and 8G12, defining a novel genotype-specific antigenic determinant on zoonotic group HEVs (Fig. 3F).

**Structural basis for the genotype-specific reactivity of 6H8**

We proceeded to experimentally confirm the importance of interface amino acids of the E2 domain involved in 6H8's specific binding through a series of structure-based mutagenesis analysis. Each of the 16 residues involved in interactions—480, 482-483, 487-490, 529-533, 559-560, and 584-585—were mutated to alanine for subsequent examination. The dimerization of these mutant proteins was verified using SDS-PAGE, and their binding activities to mAb 6H8 were assessed through enzyme-linked immunosorbent assay (ELISA). Mutations at S488A, T489A, N490A and Y532A completely abolished the binding of mAb 6H8 (Fig. 4A), highlighting the indispensable role of S488, T489, N490 and Y532 in mAb 6H8 binding. Among these four key residues, S488 and T489 are in close contact with both the heavy and light chains of mAb 6H8 (Fig. 4B, C). Y532 primarily engages with the heavy chain and forms an interaction network containing several hydrogen bonds and π-π interactions (Fig. 4D). Intriguingly, despite the absence of hydrogen bonds between the zoonotic group-specific residue N490 and mAb 6H8, its alteration led to a total loss of binding capability (Fig. 4A). The crystal structure revealed that N490 is in hydrogen bond liaison with another crucial epitope residue Y532, suggesting that it contributes to the conformational fidelity of the mAb 6H8 epitope (Fig. 4E). Consequently, conformational perturbations induced by the N490A mutation may compromise the epitope's structural integrity on E2s-4, culminating in its diminished binding to mAb 6H8, as demonstrated by the outcomes of the N490A experiment (Fig. 4A).

To investigate the role of the binding site recognized by mAb 6H8 in virus-cell attachment, we individually mutated amino acids 488, 489, 490 and 532 to alanine in both p239-1 and p239-4 VLPs to evaluate their cell-binding sensitivities. The attachment of these mutated VLPs to HepG2 and PPH cells was assessed by staining with an anti-ORF2 antibody, followed by confocal microscopy analysis to determine their binding efficiencies (Fig. S12). In PPH cells, mutations at residues 488, 490 and 532 of p239-4 completely abolished binding, while residue

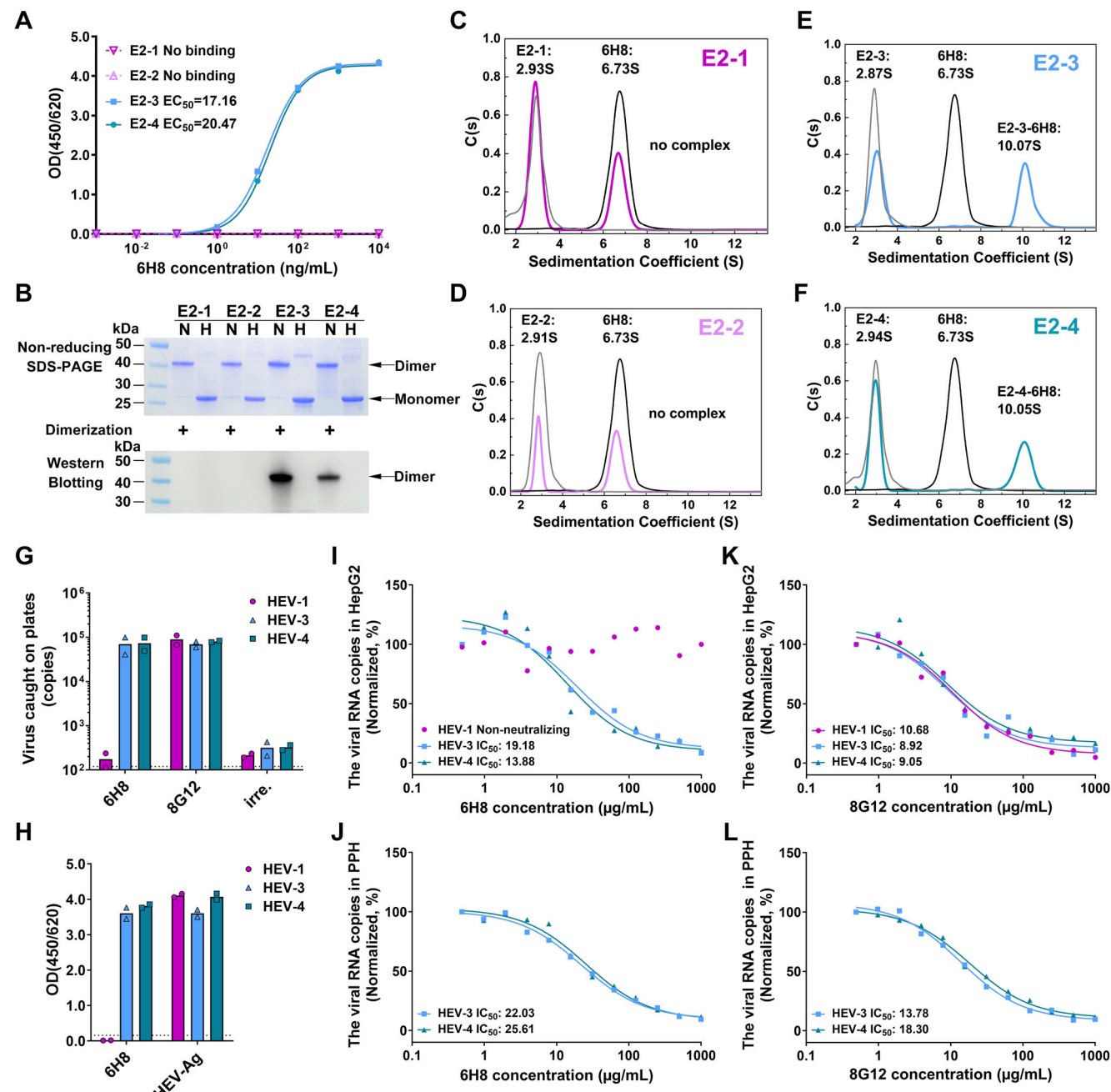

**Fig. 2 | Characterization of neutralizing mAb 6H8, which recognizes an epitope specifically present on zoonotic HEV capsid. A** Binding activity of mAb 6H8 against the four genotypes E2 proteins tested by ELISA, and EC50 was calculated using a nonlinear regression model. E2-1: genotype 1 E2; E2-2: genotype 2 E2; E2-3: genotype 3 E2; E2-4: genotype 4 E2. Data represent two biological replicates. **B** E2 proteins with four genotypes subjected to SDS-PAGE and western blotting analysis with mAb 6H8. Lanes marked with H indicate heated samples, whereas lanes marked with N indicate unheated samples. (+) denotes dimerization. **C** Sedimentation velocity used to detect mAb 6H8 binding of genotype 1 E2 (E2-1) protein. **D** Sedimentation velocity used to detect mAb 6H8 binding of genotype 2 E2 (E2-2) protein. **E** Sedimentation velocity used to detect mAb 6H8 binding of genotype 3 E2 (E2-3) protein. **F** Sedimentation velocity used to detect mAb 6H8 binding of genotype 4 E2 (E2-4) protein. **G** Different genotypes of HEV captured by mAb 6H8 in an immune capture experiment with real-time RT-PCR for viral RNA quantitation. Cross-genotype mAb 8G12 was used as a positive control, and an irrelevant mAb (irre.) was used as a negative control. Dash line indicates the limit of detection. HEV-1: genotype 1 HEV; HEV-3: genotype 3 HEV; HEV-4: genotype 4 HEV. Data represent two biological replicates. **H** Detection of different HEV genotypes captured by mAb 6H8 in an immune capture experiment with ELISA for HEV antigen quantitation. The dash line indicates the cut-off value. Data represent two biological replicates. **I** Neutralizing activity of mAb 6H8 for different genotypes of HEV in HepG2. Data represent two biological replicates. **J** Neutralizing activity of mAb 6H8 for different genotypes of HEV in primary porcine hepatocytes (PPH). Data represent two biological replicates. **K** Neutralizing activity of mAb 8G12 for different genotypes of HEV on HepG2. Data represent two biological replicates. **L** Neutralizing activity of mAb 8G12 against different genotypes of HEV in PPH. Data represent two biological replicates. mAb 8G12 with cross-genotype neutralizing capability for HEV was used as a control for the neutralizing assay. Data are presented as mean values (**A** and **G**–**L**). Source data are provided as a Source Data file.

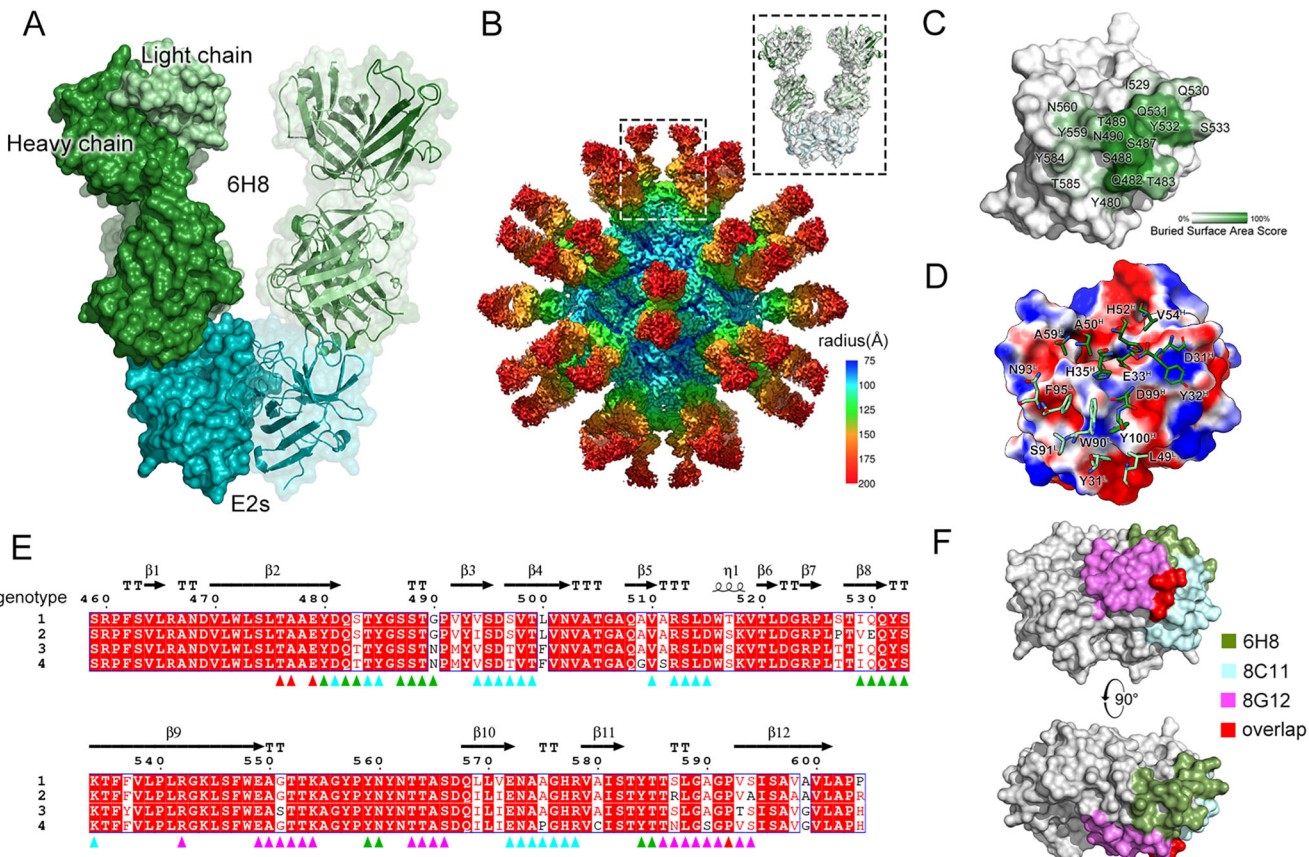

**Fig. 3 | Structure of HEV capsid protein E2s-4 in complex with the Fab of the neutralizing antibody 6H8. A** Crystal structure of the immune complex E2s-4:6H8 Fab, with E2s-4 depicted in cyan, and the heavy chain and light chain of 6H8 shown in deep green and light green, respectively. **B** Iso-contoured views (radially colored) of cryo-EM density map of genotype 4 HEV (HEV-4) virus-like particle (VLP) in complex with 6H8 Fab, with the E2s-4:6H8 Fab crystal structure well-fitted into the cryo-EM density map, as shown in the upper-right corner. **C** Surface representations of E2s-4 highlighting the interacting epitope residues, computed with the PISA program and colored according to the buried surface area score.

**D** Electrostatic potential surface of the epitope on E2s-4 (red, negative; blue, positive; and gray, neutral), with key residues for interaction from 6H8 CDR loops represented as sticks. **E** Esprit representation of a structure-based sequence alignment of E2s domains across different HEV genotypes, with secondary structural elements for E2s-1 shown on top and conserved residues highlighted in red boxes outlined in blue. Epitopes of representative antibodies (8C11, 8C12, and 6H8) designated with triangles under the alignment, with corresponding colors in Fig. 3F. **F** Surface representations of E2s exhibiting different epitopes mapped from the aforementioned three antibodies.

489 did not. In contrast, for HepG2 cells, mutations at residues 488 and 532 in both p239-1 and p239-4 disrupted binding, whereas mutations at residues 489 and 490 did not significantly affect binding compared to the wild type. Remarkably, although mutations at residue 490 in both p239-1 and p239-4 did not alter their attachment to HepG2 cells, the same mutation in p239-4 led to a marked reduction in binding to PPH cells. This indicates that residue 490 plays a crucial role not only in the specific interaction with mAb 6H8 but also in determining the viral tropism towards porcine hepatocytes.

The epitope of 6H8 is primarily located on the 480-loop (amino acids 481-492) and 530-loop (amino acids 530-534). Within these two loops, three amino acids (483, 490 and 492) differ between the human and zoonotic groups of HEVs. Specifically, in zoonotic group HEV, threonine (T), asparagine (N) and methionine (M) in residues 483, 490 and 492, are substituted by serine (S), glycine (G) and valine (V) in the human group, respectively (Fig. 3E and Fig. S11). Of these three residues, amino acid 483 was shown not to play a role in binding to 6H8 (Fig. 4A), whereas residue 490, which is crucial for both the antibody's recognition and attachment to porcine hepatocytes. Given the proximity of residue 492 to both of epitope key residues 490 and 532, we conducted a comprehensive set of reverse mutations on the E2-1 and E2-4 proteins, respectively, including both single and paired mutations specifically targeting residues 490 and 492 (Table S3). For the E2-1 mutants, altering residue 490 to asparagine (G490N) single-handedly

restored 6H8's reactivity, whereas mutating 492 to methionine (V492M) had no such effect. However, pairing these mutations (G490N/V492M) enhanced the reactivity towards 6H8 by 2.6 times more than the single G490N mutation (Table S4 and Fig. 4F). Structural analyses of both the wild-type and mutant E2s-1 proteins (E2s-1-G490N/V492M) (Table S1) revealed subtle differences in both loop regions and the positions of key residues 490, 492, and 532 (Fig. 4H). The structural conformation of the 480- and 530-loop regions (6H8 epitope region) in the structure of E2s-1-G490N/V492M is more similar to that in E2s-4 than to E2s-1, as evaluated by root mean square deviations (RMSDs) of 0.227 and 0.458 Å. A notable hydrogen bond interaction between the side chain of N490 and Y532 was observed, akin to that observed in E2s-4, which is deemed essential for maintaining the specific conformation of the 480- and 530-loops, thereby facilitating 6H8's distinctive recognition.

Similar reverse mutations were conducted on the E2s-4 protein to evaluate their impact on binding to 6H8. Consistently, only proteins mutated at N490 to glycine (N490G), whether alone or in combination with other mutations, exhibited a complete loss of reactivity towards 6H8 (Fig. 4G). Further structural analysis of the double mutant E2s-4 (E2s-4-N490G/M492V) compared to the wild-type E2s-4 revealed, as anticipated, that its conformation in the 480- and 530-loops region was more closely aligned with that of E2s-1 than with the original E2s-4 structure (Fig. 4I). Additionally, these mutations disrupted the above-

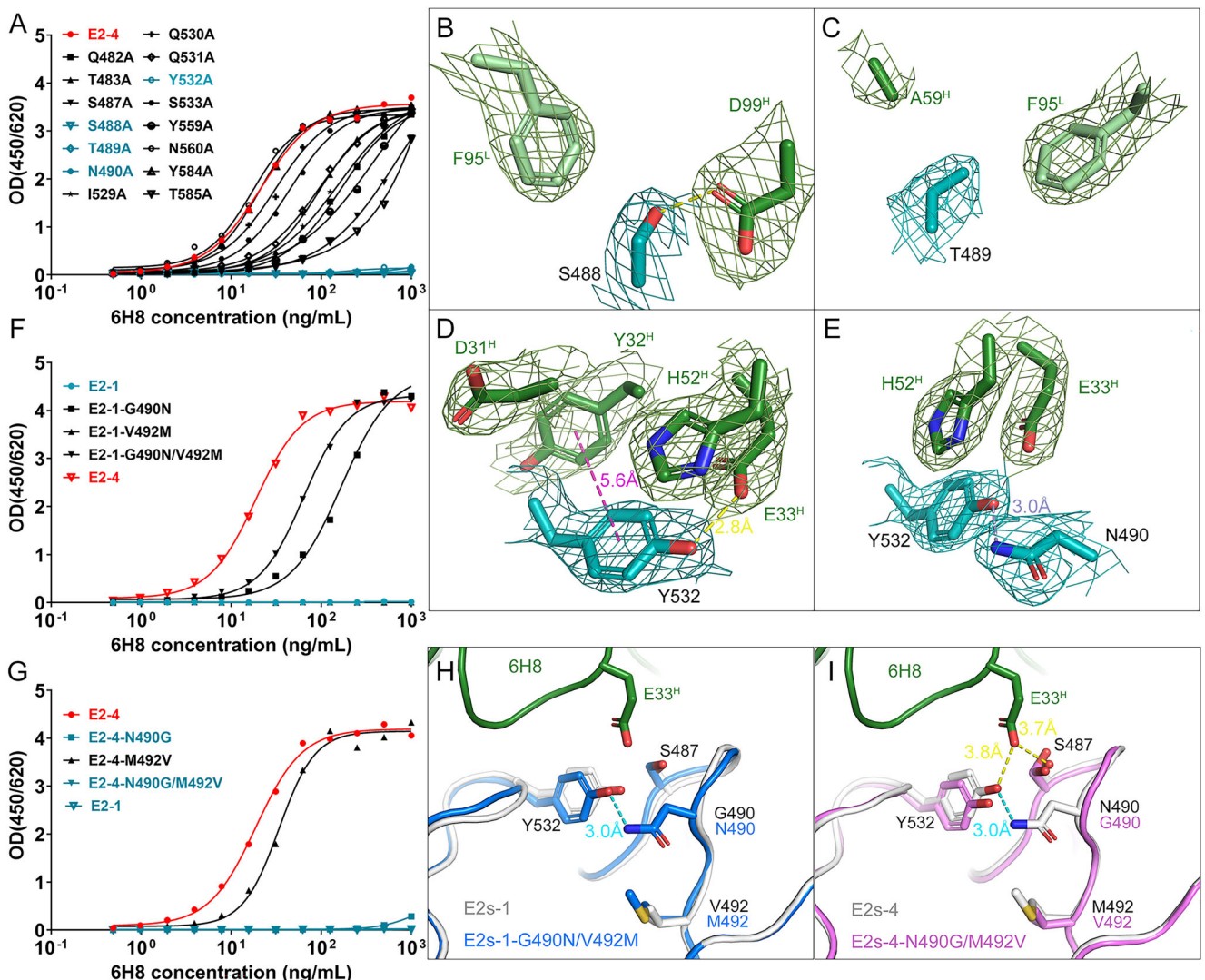

**Fig. 4 | Characterization and structural details of key residues involved in genotype 4 HEV specific reactivity of 6H8. A** Binding activity of 6H8 to genotype 4 E2 (E2-4) as well as single point mutations on epitope residues (alanine scanning). Data represent two biological replicates. **B–E** Close-up views of key epitope residues at positions 488 (**B**), 489 (**C**), 532 (**D**), and 490 (**E**), showing structural details of E2s-4:6H8 Fab interactions. Residues shown as sticks (side chain only) together with their 2Fo-Fc density map contoured at a level of 1.0σ. Residues from E2s and 6H8 colored in cyan and deep/light green (heavy chain/light chain), respectively. Intermolecular hydrogen bonds shown in yellow dashed lines, and intramolecular hydrogen bonds shown in cyan dashed lines. π-π interactions shown in magenta dashed lines. Superscripted "H" and "L" denote residues within the heavy chain and light chain, respectively. **F, G** Binding activity of 6H8 to genotype 4 E2 (E2-4) and genotype 1 E2 (E2-1) mutations (**F**), and its backward mutations (**G**). Data represent two biological replicates. **H** Superimposition of E2s-1 and E2s-1_G490N/V492M with E2s-4:6H8 complex by hiding the original E2s-4 structure, shown in cartoon representations with key residues shown as sticks. Wildtype E2s-1 depicted in white, and its G490N/V492M mutation depicted in marine. **I** Superimposition of E2s-4:6H8 and E2s-4-N490G/M492V shown in cartoon representations with key residues shown as sticks. Wildtype E2s-4 depicted in white, and its N490G/M492V mutation depicted in violet. Data are presented as mean values (**A**, **F**, and **G**). Source data are provided as a Source Data file.

mentioned hydrogen bond and π-π interaction network around residue 532 (Fig. 4I). This evidence definitively establishes residue N490 in HEV-4 as a critical determinant for the specific interaction between 6H8 and the viral capsid, with M492 playing a supportive role in maintaining the conformation critical for 6H8 epitope recognition.

## N490 and M492 play a crucial role in maintaining epitope stability of 6H8

To further investigate the impact of mutations at residues 490 and 492 on 6H8 binding across different HEV genotypes, we conducted molecular dynamics (MD) simulations using diverse E2s protein structures with different configurations at these positions. Eight resolved or modeled structures, E2s-1, E2s-1-G490N, E2s-1-V492M, E2s-1-G490N/V492M, E2s-4, E2s-4-N490G, E2s-4-M492V and E2s-4-N490G/M492V,

were subjected in 1000 nanoseconds (ns) MD simulations. The convergence of root mean square deviation (RMSD) values below 2 Å indicates sufficient structural equilibrium and suitable for further analysis (Fig. S13). To quantitatively compare the flexibility of different structures, we calculated the root mean square fluctuation (RMSF) for each residue based on the MD simulation trajectories. The RMSF profiles revealed significant structural differences between structures possessing N490 (E2s-4, E2s-1-G490N/492 M, E2s-4-M492V and E2s-1-G490N, which bind to 6H8) and those with G490 (E2s-1, E2s-1-V492M, E2s-4-N490G and E2s-4-N490G/M492V, which do not bind to 6H8) (Fig. 5A). Structures with N490 exhibit markedly lower flexibility in the 480- and 530-loop regions, which mainly constitute the 6H8 epitope, compared to structures with G490 (Fig. 5A and Fig. S14). The increased flexibility observed in the 6H8 epitope region in G490-containing

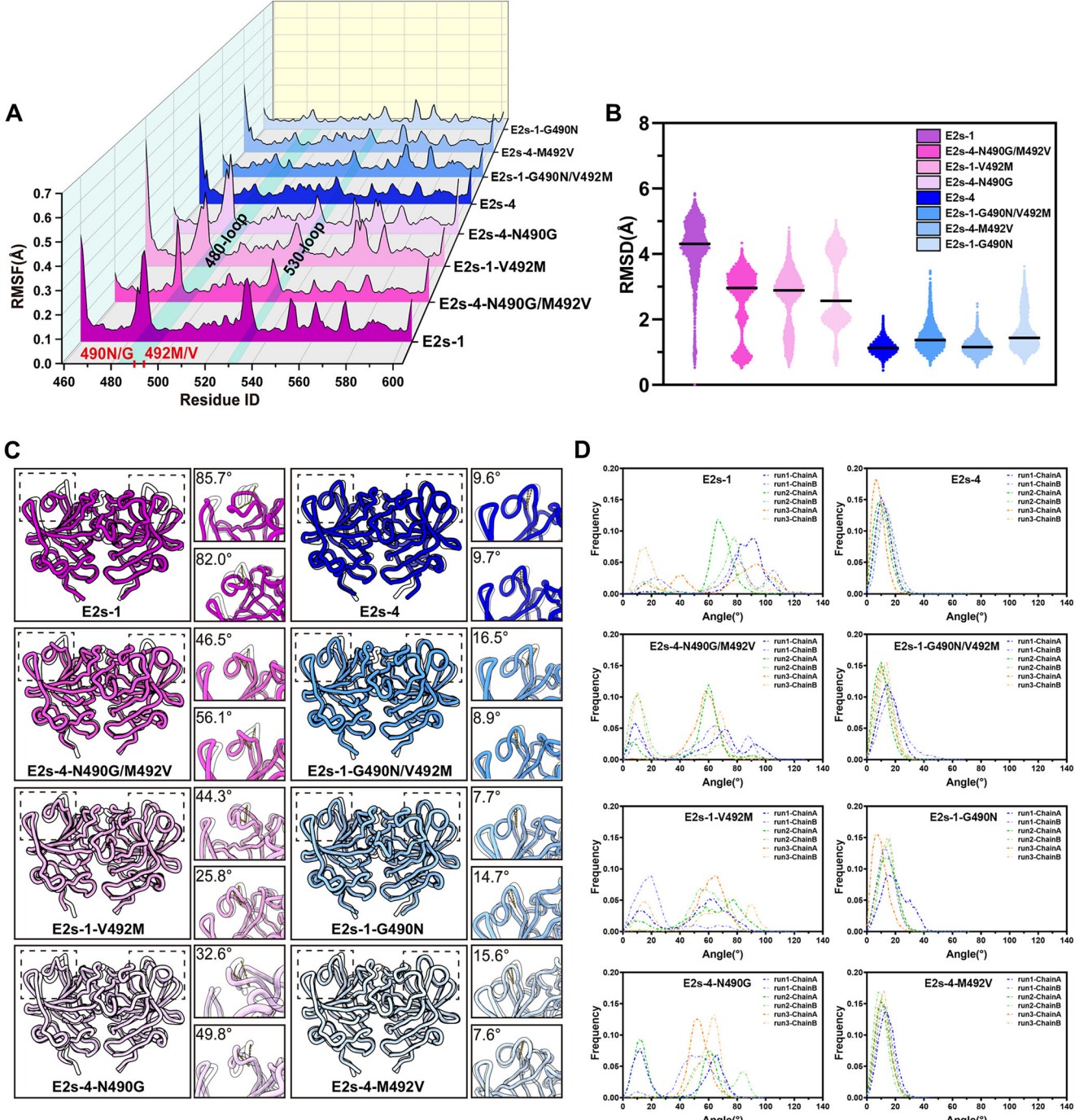

**Fig. 5 | Molecular dynamic (MD) simulations revealed the role of N490 and M492 in stabilizing 6H8 epitope. A** The root mean square fluctuation (RMSF) of various E2s structures was analyzed over 1000 nanoseconds (ns) of MD simulations. The X axis corresponds to the residue number of E2s domain, while the Y-axis shows RMSF values in Å. The two amino acids (490 and 492) and two loops (480- and 530-loops) of interest are marked in the figure. **B** Distribution of root mean square deviation (RMSD) values of the 6H8 epitope region (480- and 530-loops) derived from MD simulations. RMSD was sampled every 100 ps over the trajectory (0–1000 ns), yielding n = 10,000 frames per construct. Violin plots show the distribution of frame-wise RMSD values, with the horizontal line indicating the median. **C** The average structures of each E2s during 1000-ns MD simulations reflecting the structural deviation of the 6H8 epitope, with the black solid line denoting the initial conformation. The deflection angle of the 480-loop is defined as the angle between the Cα of residue 488 in each MD simulated frame, relative to the midpoint of Cα of residues 486 and 491 in the initial frame. **D** Distribution of deflection angles of the 480-loop region in different mutants in three replicate 1000-ns simulations. Each panel represents a distinct mutation, with six curves corresponding to three independent simulation runs (run1, run2, run3) for each chain (**A** and **B**) of the E2s dimer. E2s-1: genotype 1 E2s; E2s-4: genotype 4 E2s. Source data are provided as a Source Data file.

structures suggests that residue 490 plays a crucial role in stabilizing the epitope configuration.

To further assess the impact of residue 490 on the structure of the 6H8 epitope, we examined the structural changes of the 480- and 530-loops during MD simulation by calculating the RMSD of the backbone atoms in this region relative to their initial conformations. Structures with N490 (E2s-4, E2s-1-G490N/V492M, E2s-4-M492V, and E2s-1-G490N) maintain lower average RMSD values of 1.14 Å, 1.44 Å, 1.17 Å and 1.60 Å, respectively, indicating the stable conformations necessary for 6H8 binding (Fig. 5B and Fig. S13). In contrast, structures with G490

(E2-1, E2-4-N490G/M492V, E2-1-V492M and E2-4-N490G) exhibit significant higher deviation with average RMSD values of 4.19 Å, 2.61 Å, 2.65 Å and 2.96 Å, respectively (Fig. 5B and Fig. S13). The observed differences in the deviation of the 6H8 epitope region during MD simulations suggest that structures with G490 undergo significant conformational changes in the 6H8 epitope region, especially 480-loop, which may lead to the loss of the conformation necessary for 6H8 binding. To quantify the extent of epitope deflection, we defined the 480-loop deviation angles measured between the alpha carbon (Cα) of residue 488 in the different MD simulation frames, relative to the midpoint of the two Cα atoms of residues 486 and 491 in the initial frame. The N490-containing structures exhibits significant stability on the 480-loop during MD simulations, with average 480-loop deviation angles lower than 20°, substantially lower than those of the G490-containing structures (all greater than 25°) (Fig. 5C). Moreover, during the three replicate 1000 ns simulation periods, the G490-containing structures exhibits greater deflection in 480-loop, with significantly longer durations of marked deflection (exceeding 40°) compared to the N490-containing structures (Fig. 5D). These findings confirm that N490 plays a pivotal role in maintaining the stability of 6H8 epitope, while its substitution with glycine lead to destabilization and disruption of the epitope conformation primarily by increasing deflection in 480-loop. This evidence is consistent with our structural data, which indicate that a crucial hydrogen bond between N490 and Y532 in E2s-4, which bridges the 480- and 530-loops, is absent in G490-containing structures. Taken together, these results suggest that the loss of N490-Y532 hydrogen bond in the G490-containing structures lead to significant deflection and conformational changes in the 6H8 epitope, particularly in the 480-loop, ultimately diminishing 6H8's binding capacity.

Additionally, for the G490-containing structures, we found that E2s-1 and E2s-4-N490G/M492V, both containing V492, exhibited greater maximum degrees of deflection (greater than 100°) in the 480-loop compared to that of E2s-1-V492M and E2s-4-N490G, which contain M492 (Fig. 5D). This observation suggests that methionine at position 492 can partially mitigate the 480-loop deviations caused by the absence of asparagine at position 490. Given that N490-Y532 hydrogen bond is crucial for epitope stability, while residue 492 does not directly interact with 6H8, we analyzed the occupancy of N490-Y532 hydrogen bond over the simulation trajectory in those N490-containting structures. E2s-1-G490N/V492M and E2s-4 exhibited significantly higher occupancy rates of the N490-Y532 hydrogen bond (24.8% and 21.7%, respectively) than E2s-1-G490N (4.8%) and E2s-4-M492V (3.6%) (Fig. S15). This indicates the supportive role of M492 in maintaining the N490-Y532 hydrogen bond, thereby enhancing the binding of 6H8 to E2s.

In summary, the MD simulations revealed that N490 in HEV-4 E2s plays a crucial role in stabilizing the conformation of both 480- and 530-loops by formed an important hydrogen bond with Y532, thereby maintaining the stability and integrity of the 6H8 epitope. M492, although not as crucial as N490, also contributes to the stabilization of these loops, collectively influencing the binding affinity of 6H8.

### N490 and M492 determine HEV-4's specific binding and infection in porcine hepatocytes

The involvement of N490 and M492 in determining host tropism was further elucidated using an IFA with recombinant p239 VLPs and their mutants. In the p239-4 mutants, both p239-4-N490G and the double mutant p239-4-N490G/M492V fail to bind to PPH cells, whereas p239-4-M492V maintained the binding ability (Fig. 6A and Fig. S16). In contrast, for the p239-1 mutants, only p239-1-G490N/V492M, but not p239-1-G490N or p239-1-V492M, successfully bound to PPH cells (Fig. 6A). Fluorescence quantification further verified that although p239-4-M492V bound to PPH, demonstrating a positive signal by IFA, its fluorescence intensity was twofold lower than that of the wild-type

p239-4 (Fig. 6B). On the other hand, both p239-1-G490N/V492M and wild-type p239-4, each incorporating N490 and M492, exhibited comparable binding levels to PPH cells (Fig. 6B). Notably, all VLPs displayed similar fluorescence intensities when binding to HepG2 cells, consistent with the IFA results (Fig. 6A, C, and Fig. S17). These findings together suggest that altering asparagine at position 490 to glycine in p239-4 nullified its attachment ability to porcine hepatocytes, whereas restoring p239-1's attachment to these cells required simultaneous mutations at both positions 490 and 492 (G490N and V492M).

To further elucidate the role of residues 490 and 492 at the virus level, we rescued the 490/492 double-mutant viruses using reverse genetics technology as previously reported[34]. The wild-type and mutant HEV viruses were produced and purified from the lysate of cells transfected with the RNA of recombinant viral genomes (Fig. S18). The viral titers of HEV-1, HEV-1-G490N/V492M, HEV-4, and HEV-4-N490G/M492V were $3.32 \times 10^3$, $3.91 \times 10^3$, $2.28 \times 10^3$, and $1.91 \times 10^3$ focus-forming units (FFU)/mL, respectively (Fig. S19). The specific infectivity, defined as the RNA copy number per infectious event, was comparable among all wild-type and mutant HEV viruses, ranging from $2.56 \times 10^4$ to $5.24 \times 10^4$ copies/FFU. The binding abilities of 6H8 to both wild-type and mutant viruses were assessed using two independent methods, yielding similar results: HEV-1-G490N/V492M virus obtained the 6H8 epitope, and HEV-4-N490G/M492V virus completely destroyed the 6H8 epitope (Fig. 7A, B). These results were consistent with those obtained from p239 VLPs (Fig. 6A–C).

The binding abilities of wild-type and mutant virions to different hepatocytes were then measured using quantified real-time reverse transcription PCR (RT-PCR). As expected, all wild-type viruses and their mutants showed similar binding efficiencies with HepG2 cells (Fig. 7C). Mutant virus HEV-1-G490N/V492M and wild-type HEV-4 demonstrated equivalent binding efficacies on PPH cells (Fig. 7C). Conversely, the HEV-4-N490G/M492V virus exhibited significant reduced binding ability to PPH cells, resembling that of wild-type HEV-1 (Fig. 7C). In contrast, both HEV-4-N490G/M492V and wild-type HEV-1 viruses exhibited lower binding potency with PPH than with HepG2 cells (Fig. 7D). Nonetheless, HEV-1-G490N/V492M and wild-type HEV-4 showed similar binding potency with both HepG2 and PPH cells (Fig. 7D).

The infectivity of wild-type and mutant viruses in different hepatocytes was further assessed using ELISA- and IFA-based infection assays as previously reported[17,35]. All wild-type and their corresponding mutant viruses showed similar infectivity in HepG2 cells (Fig. 7E–H). However, HEV-1 and HEV-4-N490G/M492V showed undetectable infectivity in PPH (Fig. 7E, and Fig. 7G, H), whereas HEV-1-G490N/V492M and HEV-4 demonstrated equivalent infectivity in PPH (Fig. 7E, and Fig. 7G, H). Both HEV-1-G490N/V492M and HEV-4 displayed similar infectivity in PPH and HepG2 cells, while HEV-1 and HEV-4-N490G/M492V showed reduced infectivity in PPH compared to HepG2 cells (Fig. 7F). All wild-type and mutant viruses in PPH and HepG2 could be neutralized by mAb A286 (Fig. 7E, G, Fig. S3, and Fig. S4). In contrast, mAb 6H8 effectively neutralized HEV-4 and HEV-1-G490N/V492M in both cell types, but failed to neutralize HEV-1 and HEV-4-N490G/M492V in HepG2 cells (Fig. 7E, G, Fig. S3, and Fig. S4).

Collectively, these findings imply that the G490N and V492M substitutions in the HEV-1 virus lead to a loss of genotype-specific antigenic determinant and tropism for porcine hepatocytes.

### N490 and M492 are crucial but not the sole factors for HEV-4 infection in miniature swine

Although HEV-1-G490N/V492M virus retained binding abilities to porcine hepatocytes in vitro and HEV-4-N490G/M492V exhibited a loss of this capacity, the infectivity of these two mutant viruses in swine in vivo remains undetermined. To assess the impact of N490 and M492 on cross-species infectivity, experiments were conducted with both wild-type and mutant HEVs in both miniature swine (Bama miniature

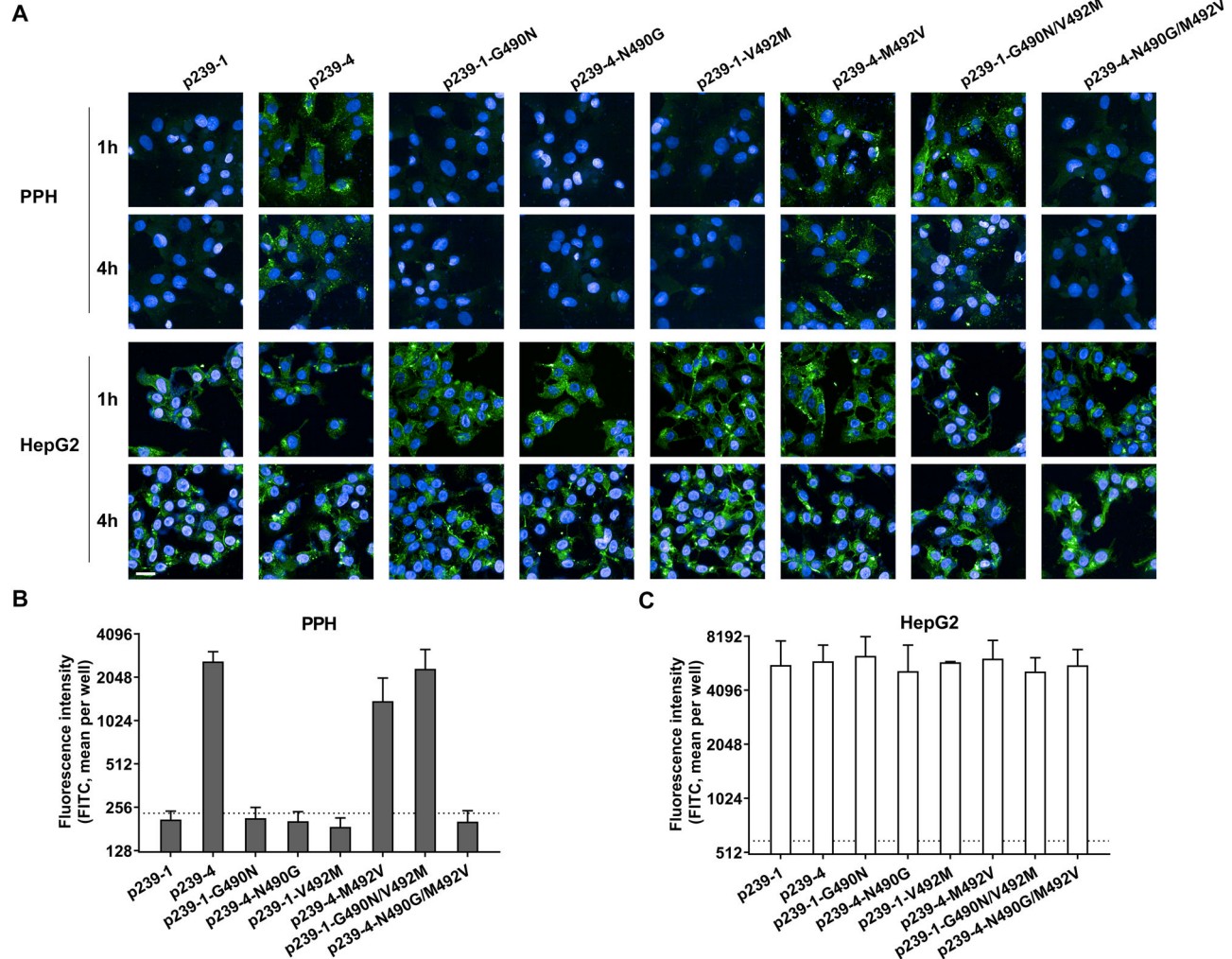

**Fig. 6 | The critical role of amino acids 490 and 492 in HEV binding to swine liver cells. A** Immunofluorescence analysis showing the binding (1 h) and penetration (4 h) of genotype 1 p239 (p239-1) and genotype 4 p239 (p239-4) and the mutant virus-like particles (VLPs) on primary porcine hepatocytes (PPH) and HepG2 cells. Green represents positive staining for p239 protein, and blue represents 4′,6-diamidino-2-phenylindole (DAPI) nuclear staining. Scale bar, 25 µm. **B** Analysis of intracellular fluorescence intensity at 1 h after p239s incubation with PPH, detected by the Operetta CLS High-Content analysis system (PerkinElmer). The dash line indicates the cut-off value. The following numbers of cells were counted: p239-1 ($n = 2299$), p239-4 ($n = 2392$), p239-1-G490N ($n = 2527$), p239-4-N490G ($n = 2064$,

p239-1-V492M ($n = 2210$), p239-4-M492V ($n = 2083$), p239-1-G490N/V492M ($n = 2334$), and p239-4-N490G/M492V ($n = 2359$). **C** Analysis of intracellular fluorescence intensity at 1 h after p239s incubation with HepG2 cells, detected by the Operetta CLS High-Content analysis system. The dash line indicates the cut-off value. The following numbers of cell counts were counted: p239-1 ($n = 2327$), p239-4 ($n = 2114$), p239-1-G490N ($n = 2333$), p239-4-N490G ($n = 1904$), p239-1-V492M ($n = 1996$), p239-4-M492V ($n = 1999$), p239-1-G490N/V492M ($n = 2434$), and p239-4-N490G/M492V ($n = 1845$). Data are presented as mean ± SD (**B**, **C**). Source data are provided as a Source Data file.

swine) and cynomolgus monkeys (*Macaca fascicularis*), respectively, following previously established protocols[18,31,36,37]. The experimental design includes four groups of cynomolgus monkeys infected with either HEV-1, HEV-4, HEV-1-G490N/V492M or HEV-4-N490G/M492V, alongside four corresponding groups of miniature swine (Fig. 7I, J). After infection, stool and serum samples were collected biweekly to detect HEV RNA via RT-PCR and to measure total anti-HEV antibodies in serum. All monkeys exhibited obvious HEV viremia, with onset between 1.0- and 1.5-weeks post-infection, persisting for 2.5-5.0 weeks (Fig. 7I). Fecal virus shedding commenced 0.5 weeks post-infection and ceased between 3.5-6.5 weeks (Fig. 7I). Anti-HEV antibodies were detectable in all groups at 1.5-3.0 weeks post-infection and remained detectable throughout the study (Fig. 7I). These findings demonstrate that both wild-type and mutant HEVs effectively infect cynomolgus monkeys, indicating that the mutations at residues 490 and 492 do not compromise the viruses' infective capabilities in non-human primates.

Bama miniature swine have been shown to be susceptible to HEV-3 and HEV-4, but not to HEV-1[37]. In our tests, only those Bama miniature swine infected with wild-type HEV-4 exhibited clear signs of HEV viremia, viral shedding, and anti-HEV antibody responses (Fig. 7J). The duration of viremia and viral shedding was ~5.5−6.0 weeks and 5.0−6.0 weeks, respectively. The anti-HEV antibody responses in swine infected with HEV-4 appeared later and lasted for a shorter duration compared to those in monkeys infected with HEV-4. Of note, the groups exposed to HEV-1, HEV-4-N490G/M492V and HEV-1-G490N/V492M exhibited no detectable fecal virus shedding, HEV viremia, or anti-HEV antibody seroconversion throughout the study, indicating a lack of infection by these viruses in swine (Fig. 7J). The diminished infectivity of the HEV-4-N490G/M492V mutant in swine underscores the pivotal role of residues N490 and M492 in HEV-4's capability to infect swine. Contrary to expectations, the HEV-1-G490N/V492M mutant, despite regaining

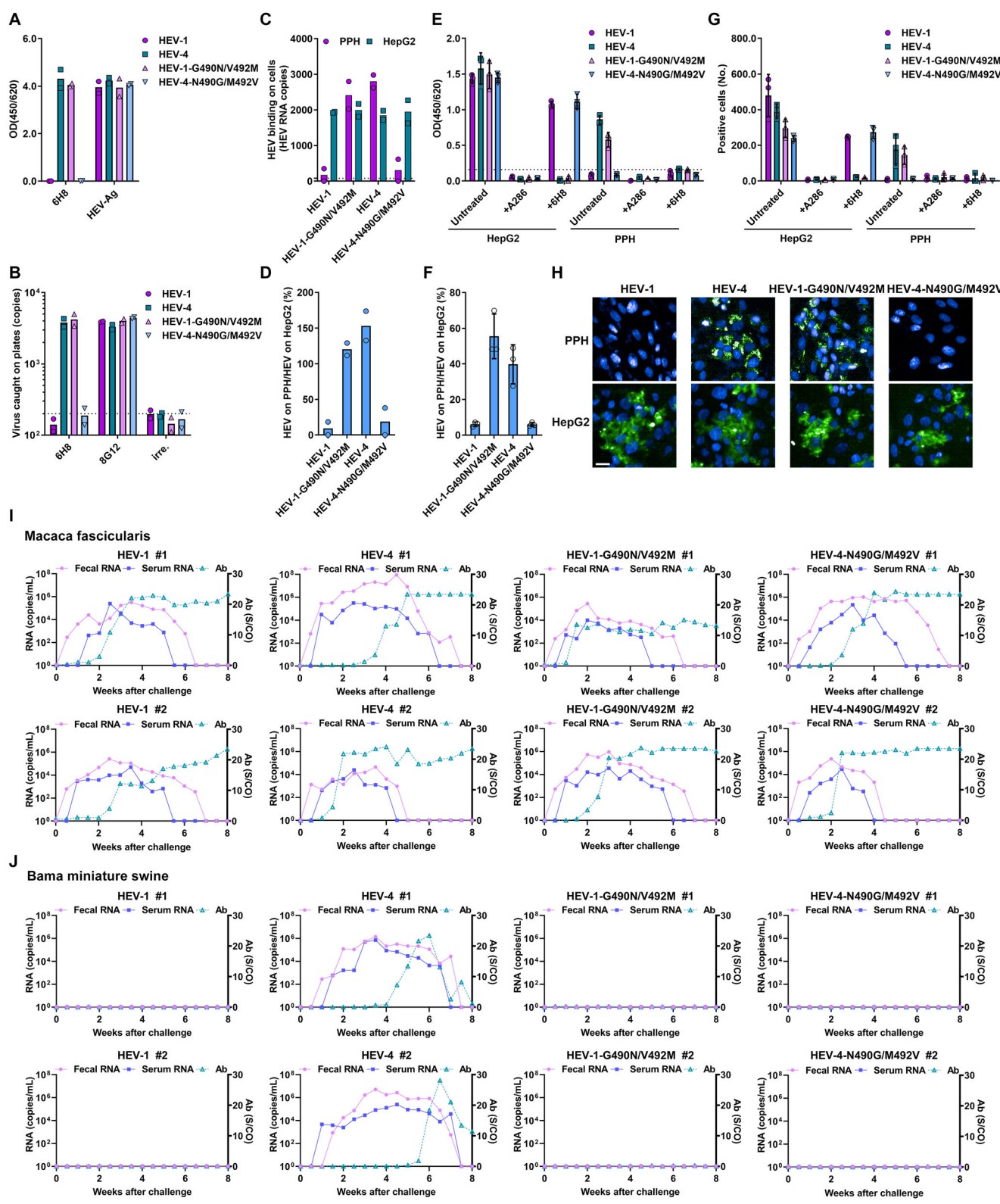

binding affinity and infectivity on porcine hepatocytes (Fig. 7C, E, G, H), failed to initiate infection in Bama miniature swine (Fig. 7I), suggesting that altering residues 490 and 492 alone is insufficient to reestablish HEV-1 infection in this species. These findings suggest that additional factors, beyond residues 490 and 492, may collectively influence HEV-4's specific tropism for swine. Overall, this study emphasizes that while residues 490 and 492 play a critical role in HEV-4 infectivity in swine, they are not solely decisive.

## Discussion

Cross-species virus transmission poses a significant global health risk, fueled by ubiquitous animal reservoirs that facilitate the proliferation and mutation of these viruses. Over the past two decades, the world has witnessed two major pandemics: the 2009 H1N1 influenza pandemic, originating from swine virus[38], and the Coronavirus Disease 2019 (COVID-19) pandemic caused by severe acute respiratory syndrome coronavirus 2 (SARS-CoV-2)[39]. Both these pandemics resulted

**Fig. 7 | The critical role of amino acids 490 and 492 in HEV infection in primary porcine hepatocytes in vitro and in swine in vivo. A** Detection of HEV ORF2 protein in supernatants of mutated HEV RNA transfected S10-3 cells. Data represent two biological replicates. **B** Mutated HEV captured by mAb 6H8 in an immune capture experiment with real time reverse transcription PCR for viral RNA quantitation. Data represent two biological replicates. Cross-genotype mAb 8G12 and an irrelevant mAb (irre.) were used as the positive and negative control, respectively. **C** Binding capability of the mutant viruses in PPH and HepG2 cells detected by real-time reverse transcription PCR. The dash line indicates the limit of quantitation. Data represent two biological replicates. **D** The ratio of HEV RNA copies bound to PPH and HepG2 cells. Data represent two biological replicates. **E** Infectivity of wild-type and mutant viruses in PPH and HepG2 cells detected by ELISA for HEV ORF2 level. Data represent three biological replicates. +A286 and +6H8 represent the infectivity of HEV in the presence of mAb A286 and mAb 6H8, respectively. **F** Ratio of HEV levels in the supernatant of PPH and HepG2 cells after HEV infection. Data represent three biological replicates. **G** Number of positive cells detected by immunofluorescence assay after HEV infection. -69 imaging areas were quantified. Data represent three biological replicates. +A286 and +6H8 represent the number of positive cells detected by immunofluorescence assay after HEV infection in the presence of mAb A286 and mAb 6H8, respectively. **H** Infection capability of mutant viruses in PPH and HepG2 cells detected by immunofluorescence assay. Green fluorescence indicates HEV ORF2 protein expression and blue fluorescence indicates positive for 4′,6-diamidino-2-phenylindole (DAPI) nuclear staining. Scale bar, 25 μm. **I** Detection of HEV infection markers in *Macaca fascicularis* inoculated with wildtype and mutant viruses. A signal to cut-off (S/CO) value of ≥1 was defined as positive. Fecal RNA: HEV RNA in the stool; Serum RNA: HEV RNA in the serum; Ab: anti-HEV antibody. **J** Detection of HEV infection markers in Bama miniature swine inoculated with wildtype and mutant viruses. A S/CO value of ≥1 was defined as positive. Fecal RNA: HEV RNA in the stool; Serum RNA: HEV RNA in the serum; Ab: anti-HEV antibody. HEV-1: genotype 1 HEV; HEV-4: genotype 4 HEV; PPH: primary porcine hepatocytes. Data are presented as mean values (**A–D**) or as mean ± SD (**E–G**). Source data are provided as a Source Data file.

from cross-species transmission[38]. Therefore, understanding the molecular mechanisms underlying such transmission is vital for developing effective prevention and control strategies. HEV, a leading cause of viral hepatitis in humans, exhibits distinct host tropism across different genotypes[40]. Some HEVs are human and non-human primates specific, while others are zoonotic infective in nature, possessing a broader host range and the ability to cross species barriers[40]. Investigating the zoonotic transmission of HEV and its mechanisms will enhance our understanding of HEV and other hepatitis viruses, as well as the evolutionary processes driving cross-species transmission.

Over the past decade, significant progress has been made in identifying the viral determinants involved in HEV's cross-species transmission[17–21,25,41,42]. Nevertheless, the precise mechanism underlying HEV cross-species transmission remain elusive. ORF2 is an important candidate for affecting cross-species transmission, but its role in cross-species transmission remains controversial[18–20,25,42]. Given its critical role in cell attachment and infection, the capsid protein of HEV (encoded by ORF2) is speculated to be an important determinant of HEV host range[25,42]. Previous studies have demonstrated the crucial role of HEV capsid protein in determining host range in vitro[24,26]. Meanwhile, other researches indicated that the ORF2 protein does not play a role in cross-species infection of HEV, as chimeric viruses with the ORF2 gene from HEV-3 or HEV-4 integrated into the HEV-1 genome failed to establish infection in swine or broaden the host range[18,41]. In this study, chimeric viruses incorporating zoonotic N490 and M492 within the backbone of HEV-1 genome (HEV-1-G490N/V492M), although restore binding and infection abilities in porcine hepatocytes in vitro, were proven fail to infect Bama miniature swine in vivo. However, HEV-4 based mutant viruses in which the zoonotic-specific determinant of ORF2 was altered by swapping residues 490 and 492 (HEV-4-N490G/M492V), completely loss the infectivity to Bama miniature swine, suggesting a significant but insufficient role for ORF2 in dictating HEV host specificity, with residues 490 and 492 as critical amino acids. The failure of modified HEV-1 with N490 and M492 to infect Bama miniature swine implies that, alongside ORF2, additional genomic regions are crucial in defining HEV's host preferences. This is supported by earlier inquiries into the roles of ORF1[18–20], 5′ NCR[18], codon usage[21,43], and adaptive evolution[17,22,23] in HEV's host tropism and cross-species transmission. ORF1 plays a significant but insufficient role in HEV host range determination, as chimeric viruses HEV-1 with HEV-4 or HEV-3 ORF1 could replicate in porcine kidney cells yet failed to infect pigs[19,20]. Insertion of human ribosome protein sequence S17 into ORF1 expanded the host range in cultured cells but failed to expand HEV host tropism in vivo[20]. Critical amino acids within ORF1 and ORF2 may cooperatively determine HEV host tropism[22,23]. Notably, none of the viral factors associated with HEV host range have been shown to restore viral infectivity in vivo. Beyond viral factors, host determinants, such as host cellular factors and host immune status,

may also contribute to HEV host tropism[17,34,41]. Collectively, these observations underscore the complexity of the mechanism underlying HEV host specificity. Emerging technologies, such as the lipid nanoparticle-encapsulated viral RNA platform[44], may offer may offer more approaches.

In this study, we utilized a zoonotic group HEV-specific mAb 6H8 to precisely identify a neutralizing epitope on pORF2 which is also critical for recognizing porcine hepatocytes and infecting swine. Notably, the 6H8 epitope's location has been demonstrated as critical for binding to the cellular receptor for HEV[45]. The residues S488, T489, N490 and Y532 are vital for the 6H8 epitope's integrity, with residues 490 and 492 crucial for distinguishing zoonotic from human group HEV. This mirrors findings from prior research that identified residue 512 as key in the HEV-1-specific 8C11 neutralizing epitope, and residue 497 significant for genotype identification by 8C11[33]. Structurally, residues N490 and M492 in zoonotic group HEV have elongated side chains compared to those in human group HEV, enhancing the interaction between N490 and Y532. This interaction is pivotal for maintaining the spatial conformation between two loops forming the domain and mediating receptor engagement. Although crystal structures reveal only subtle differences between these two loops in both HEV groups, which may result from crystallization techniques, cryo-EM observations of HEV p495 VLP highlight the instability of this domain in human group HEV[46]. We proposed that the epitope's stability may be fundamental for zoonotic HEV's binding to porcine hepatocyte cells. MD simulations confirmed that N490 in HEV-4 E2s indeed plays a critical role in stabilizing the conformation of both 480- and 530-loops by formed an important hydrogen bond with Y532 and M492 also contributes to the stabilization of these loops, thereby collectively influencing the binding affinity of 6H8. The difference in the stability of this domain in human and zoonotic HEV groups may also influence viral attachment to PPH cell and infection in swine.

A previous study showed that HEV-1 and HEV-3 exhibit different binding pattern with swine kidney cells[24]. In this study, we found that HEV-1 and HEV-4 display distinct binding and entry capabilities in porcine hepatocytes. These suggested that the cellular molecules involved in different genotypes of HEV in attachment and entry into PPH may differ. p239-1 could only partially block the binding of HEV-4 on HepG2 cells and could not block the binding of HEV-4 on PPH, suggesting that there may be a unique HEV-4-specific cellular receptor or co-receptor mediating HEV-4 infection. Similar findings were reported in a previous study[24], which indicated that the receptors (or co receptors) for HEV-1 and HEV-3 on HepG2/C3A are different. The epidermal growth factor receptor (EGFR), integrin alpha 3 (ITGA3), and heparan sulfate proteoglycans (HSPGs) have been identified as key molecules involved in HEV attachment and/or entry[42,47–49]. These studies, however, were all conducted on human cells using a single genotype of HEV (either HEV-1 or HEV-3)[42,47,48]. Further research is needed

to determine whether there are differences in the roles of these molecules in the cell attachment and entry of different HEV genotypes.

All four major genotypes of HEV (HEV-1 to HEV-4) infecting human belong to a single serotype and a cross-genotype neutralizing epitope recognized by 8G12 had been identified[31]. However, a study revealed that there were differences in immunogenicity between HEV-1 and HEV-4 ORF2s[50]. Multiple studies had also identified HEV genotype-specific or group-specific neutralizing epitopes on ORF2, including the HEV-1-specific epitope recognized by 8C11, the human group HEV-specific epitope recognized by 2B1, and the zoonotic group HEV-specific epitope recognized by 4C5[33,50]. Our previous study suggested that the epitopes recognized by 6H8 and 8C11 contribute to the formation of strong immunodominant epitopes on the E2s domain[32]. These specific epitopes may underlie the differences in the immunogenicity of ORF2s between HEV-1 and HEV-4. Given the different epidemiology and clinical features of human group and zoonotic groups HEV[27], these findings may help to differentiate the source of HEV infection and contribute to the development of precise prevention and control strategies for different groups or genotypes of HEV.

In this study, we found that two amino acid mutations, G490N and V492M, in HEV-1 altered its cell tropism and enable binding to PPH, highlighting the critical nature of residues 490 and 492 in facilitating HEV cross-species infection. Mutations of these two residues may expand HEV's host range, increasing its transmissibility. Similar phenomena have been observed in many other viruses. For instance, two amino acid substitutions in the Spike (S) proteins significantly influenced human adaptation in both severe acute respiratory syndrome coronavirus (SARS-CoV) and middle east respiratory syndrome CoV (MERS-CoV)[51,52]. Likewise, in MERS-CoV research, a pair of amino acid alterations allowed a bat coronavirus (HKU4) to infect human cells[52]. Furthermore, a single mutation at position 82 in the glycoprotein of Ebola viruses, and a single mutation at position 372 in the S protein of SARS-CoV-2, has been implicated in facilitating human infection[53–55]. These examples illustrate how minimal genetic alterations can significantly shift a virus's host adaptability.

This study had some limitations. First, PPH, primary cells isolated from natural host of HEV, rather than stable porcine cell lines, were primarily used in this study. Compared with stable cells lines, PPH may exhibit potential variability between preparations. However, the isolation and culture of PPH have been well-established in numerous studies[28,29,56–59]. Moreover, PPH have been successfully utilized as a model system for HEV research in previous investigations[28,29]. Second, although the animal experiments were conducted as previously described[18,31,37], the sample size was relatively small in each experimental group. Nevertheless, animals in each group exhibited consistent results, supporting the validity of our findings.

In summary, our study pinpointed a zoonotic group HEV-specific neutralizing epitope on pORF2 that is targeted by mAb 6H8. We found that the residues 490 and 492, crucial for 6H8 recognition and differ between human and zoonotic HEV groups, are instrumental in mediating the attachment of HEV-4 capsids and virions to porcine hepatocytes and subsequent infection of swine by zoonotic group HEV. Our findings offer significant insights into the molecular mechanisms driving host tropism disparities between human and zoonotic HEVs, highlighting a potential avenue for mitigating cross-species HEV transmission.

## Methods
### Ethics statement
Primary human hepatocytes were obtained under ethical approval from the China Ethics Committee of Registering Clinical Trials (approval number: ChiECRCT20210471). Donor identities were blinded to the researchers by using anonymized sample identifiers, and all studies were conducted using de-identified samples. All animal experiments were approved by the Institutional Animal Care and Use Committee and Laboratory Animal Management Ethics Committee of Xiamen University. All experiments were performed in accordance with the relevant guidelines and regulations of Xiamen University.

### Cell lines
HepG2 (HB-8065) cells were obtained from the American Type Culture Collection (ATCC, MD, USA) and maintained in Dulbecco's modified Eagle's medium (DMEM) (Gibco, CA, USA) supplemented with 10% fetal bovine serum (FBS) (GIBCO, CA, USA).

### Viruses, antibodies, and proteins
The genotype 1 virus (strain Xinjiang, GenBank accession no. L08816.1), genotype 3 virus (strain JRC-HE3, GenBank: accession no. AB630971.1), and genotype 4 virus (strain Ch-S-1, GenBank: accession no. EF077630) were isolated from stool samples obtained from rhesus monkeys challenged with HEV. The stool samples containing virus were suspended in 10% weight /volume (w/v) of phosphate-buffered saline (PBS) containing 1% w/v bovine serum albumin (BSA). The suspension was centrifuged at 10,000×g for 20 min, and the supernatant was harvested, filtered with a 0.22 μm membrane filter, and stored at -20 °C until use. HEV-1 strain Sar-55 (GenBank accession no. AF444002) and HEV-4 strain D32 (GenBank accession no. PP475173.1) were generated from infectious cDNA clones by plasmid-based reverse genetics technology as described previously with some modifications[34]. The mAb 6H8 was screened using a murine mAb preparation protocol as previously reported[32,60]. The E2 (amino acids 394-606), E2s (amino acids 455–602) and p239 (amino acids 368-606) proteins were recombinantly expressed and purified as previously described[61]. A series of mutants were generated by site-directed mutagenesis using PCR and expressed following the same procedure as the wild-type proteins[32].

### Primary Porcine Hepatocyte Isolation and Culture
The isolation and culture of PPH were performed using a modified two-step method of collagenase perfusion followed by differential centrifugation[28,29,58]. In brief, the liver of miniature swine was perfused with perfusion solutions containing collagenase via the portal vein. Hepatocytes were dispersed and centrifuged in medium three times at $100 \times g$. After centrifugation, the hepatocytes were resuspended and seeded in collagen-coated culture plates in William's medium E (containing 1 μM dexamethasone, 0.1 mM insulin, 20 ng/mL epidermal growth factor (EGF), 20 ng/mL fibroblast growth factor (FGF), 20 ng/mL hepatocyte growth factor (HGF), and 10% FBS and maintained at 37 °C in the presence of 5% $CO_2$.

### Primary Human Hepatocyte Culture
Cryopreserved PHH were obtained from Liwo Biotechnology (Shanghai, China) and seeded at a density of $1 \times 10^6$ viable cells per well in collagen-coated 6-well plates (BD Biosciences, CA, USA). After 4 h-attachment at 37 °C with 5% $CO_2$, cells were maintained in DMEM/F12 medium (Gibco, CA, USA) supplemented with the following components: 2% B27 supplement minus vitamin A (Shanghai Yuanye, Shanghai, China), 1 mM N-acetyl-L-cysteine (Sigma-Aldrich, MO, USA), 10 mM nicotinamide (Sigma-Aldrich, MO, USA), 1% L-alanyl-L-glutamine dipeptide (GlutaMAX), 1% penicillin, 1% streptomycin. Growth factors were added at the following concentrations: 50 ng/mL EGF (Sinobiological, Beijing, China), 100 ng/mL FGF10 (Sinobiological, Beijing, China), 25 ng/mL HGF (Sinobiological, Beijing, China), and 50 ng/mL FGF7 (Thermo Fisher Scientific). Small-molecule inhibitors were also included: 10 μM Y-27632 ROCK inhibitor (Enzo Life Sciences, New York, USA), 100 nM A83-01 (TopScience, Shanghai, China), and 3 μM CHIR-99021 (TopScience, Shanghai, China). Cultures were maintained at 37 °C in a humidified with 5% $CO_2$, and the medium was changed daily.

## Virus Rescue

The infectious cDNA clone of HEV-1 strain Sar-55, pSK-HEV-2 (GenBank accession no. AF444002), was donated by Emerson[34]. HEV-4 strain D32 (GenBank accession no. PP475173.1) was isolated from the bile of swine in a slaughterhouse in Xiamen, China. Standard molecular techniques were used to assemble a full-length cDNA clone of the D32 strain from cDNA fragments that were produced by RT-PCR. HEV-4-N490G/M492V was generated in the backbone of D32 by replacing the residues asparagine at 490 and methionine at 492 of ORF2 with glycine and valine, respectively. And HEV-1-G490N/V492M was generated in the backbone of Sar-55 by replacing the residues glycine at 490 and valine at 492 of ORF2 with asparagine and methionine, respectively. The wild-type and mutant HEV stocks were produced by plasmid-based reverse genetics technology as described previously with some modifications[34]. Briefly, plasmid pSK-HEV-2 and D32 were both linearized with the restriction enzyme *BglII* downstream of the poly(A) tail of HEV. In vitro transcribed capped RNA transcripts were produced with the mMESSAGE mMACHINE T7 Transcription Kit (Invitrogen, CA, USA) at 20 °C for 16 h using 2 μg of linearized DNA. In vitro-transcribed capped RNA transcripts were then transfected into S10-3 cells (a subclone of human hepatoma Huh-7 cells). The transfected cells were cultured at 37 °C in DMEM containing 10% FBS for 15 to 20 days and then lysed in distilled $H_2O$ by three repeated cycles of freeze and thaw. After centrifugation for 30 min at $10,000 \times g$ to remove cell debris, the supernatant was harvested as HEV stock. Viral titers were quantified by real-time RT-PCR as described below. To quantify HEV infectious titers, HepG2/C3A cells were seeded in 96-well plates at $2 \times 10^4$ cells per well one day prior to the infection assay. Cells were infected with viruses for 8 h. Five days post-infection, cells were analyzed by immuno-fluorescence assay as described below. The number of FFU was counted according to a previous report[28].

## Real-time reverse transcription PCR

Viral RNA levels were quantified using real-time RT-PCR as previously reported[62]. Real-time RT-PCR was carried out using a commercial One-Step RT-PCR Kit (Genmagbio, Beijing, China) on a CFX96TM Real-Time System and C1000TM thermal cycler device (Bio-Rad, CA, USA). Plasmids with known viral copy numbers were serially diluted to generate standard curves for HEV RNA quantification.

## SPR assay

The binding affinities of mAb 6H8 for different genotype E2 proteins were measured by SPR on BIAcore 3000 (GE Healthcare, USA) at 25 °C in HBS (10 mM HEPES, pH 7.4, and 150 mM NaCl). One channel of a CM-5 sensor chip was coated with 290 response units (RU) of mAb 6H8, and the other flow cell was left uncoated and blocked as a reference channel. Serially diluted antigen was flowed through the mAb 6H8-coated channel and the reference channel at 30 μL/min for 200 s followed by a 700-second dissociation phase. Sensorgrams of serial concentrations were globally fitted using a 1:1 Langmuir model.

## Indirect ELISA

The antigen (100 ng per well) was coated onto 96-well microplates and incubated at 37 °C for 2 h. After blocking with blocking reagent (PBS containing 2% BSA), the antigen-coated microplates were washed once with wash buffer (PBS with 0.5% Tween-20, PBST). Serially 10-fold diluted mAbs were added to the antigen-coated plates and incubated with antigen at 37 °C for 30 min. After five washes with PBST buffer, the bound antibodies were detected with horseradish peroxidase (HRP)-conjugated goat anti-mouse (GAM) antibody at 37 °C for 30 min. Subsequently, the plates were washed five times and the color was developed with 100 μL of tetramethylbenzidine (TMB) substrate. The optical density (OD) was examined with a microplate reader (Autobio, Zhengzhou, China) at 450 nm with a reference of 620 nm.

## Western blotting

Western blotting was conducted in accordance with previously described methods[60]. Briefly, recombinant E2 proteins with or without boiling were electrophoresed on SDS-PAGE gel and subsequently transferred to nitrocellulose membranes (Fig. 2B). The membranes were blocked with 5% skim milk in PBS for 30 min and washed with PBST. Subsequently, the membranes were incubated sequentially with mAb 6H8 followed by HRP-conjugated GAM antibody. The color was developed using SuperSignal West Femto Maximum Sensitivity Substrate (PIERCE, MA, USA) and the reaction signals were determined using Image Quant LAS4000mini (GE Healthcare, Little Chalfont, UK).

## Analytical ultracentrifugation

Sedimentation velocity experiments were performed at 20 °C on a Beckman XL-A AUC, equipped with absorbance optics and an An60-Ti rotor, as previously described[63]. Briefly, all samples were diluted to ~1.0 $OD_{280}$ in a buffer containing 20 mM HEPES (pH 7.5) and 150 mM NaCl, in a 1.2-cm light length cell. The rotor speed was set at 40,000 rpm. E2 proteins and mAb 6H8 were pre-incubated at a molar ratio of 2:1 at 37 °C for 2 h to allow for any possible interactions. Sedimentation coefficient values were obtained using the c(s) method with the Sedfit software kindly provided by Dr. Schuck P (National Institutes of Health, MA, USA).

## Immune capture assay

The immune capture assay was performed based on real-time RT-PCR assay and ELISA as previously described[32,64]. The plates were coated with 900 ng of mAbs per well. After a single wash, the antibody-coated plates were blocked with 350 μL of blocking reagent for 2 h at 37 °C. Then, the plates were washed once and incubated with a 200-μL aliquot of HEV suspension at 37 °C for 2 h. For the real-time RT-PCR assay, after rinsing five times with wash buffer, 200 μL of TRIzol was added to each well and incubated at 4 °C for 10 min. HEV RNA was purified, and the viral RNA levels captured by the mAbs were measured with a real-time RT-PCR assay as described above. For the ELISA assay, plates were rinsed five times with wash buffer, and 200 μL of HRP-conjugated mAb no.4 solution was added to each well, followed by incubation at 37 °C for 30 min. After five additional washes, 100 μL of TMB substrate solution was added to each well and incubated for 15 min at 37 °C. Absorbance was measured using a microplate reader.

## Detection of HEV antigen by ELISA

The antigen in the cell supernatants was detected using a commercial ELISA kit (Wantai, Beijing, China) and an in-house sandwich ELISA. The antigen detection kits were developed with mAb 12F12 as the capture antibody and mAb no.4 as the detection antibody[64] and can be used for research purposes only. The detection of HEV antigen using the commercial ELISA kit was performed according to the manufacturer's instructions.

The protocol for the in-house sandwich ELISA was the same as described previously[64], except that mAb 6H8 was used as the capture antibody instead of mAb 12F12. Briefly, 96-well ELISA plates were coated with mAb 6H8 (500 ng per well) and incubated overnight at 4 °C. The plates were washed once with PBST and subsequently blocked with blocking reagent. Samples (50 μL) were then added to the wells and incubated at 37 °C for 60 min. A 100-μL aliquot of no. 4-HRP solution was added directly to each well without washing. Following an additional 30 min incubation, the plates were washed five times with PBST. Color development and OD measurement were performed as described above.

## HEV cell-binding assay

HepG2 cells, PPH, and PHH were seeded into 96-well plates or 6-well plates and then treated with one of the following three assays. For the binding assay, serially 2-fold-diluted (initial 64 copies per cell) HEV

virions were directly added to both HepG2 cells and PPH or 1000 copies per cell HEV virions were incubated with PHH. In the p239 competition infection assay, both HepG2 cells and PPH were pre-incubated with serial 2-fold dilutions of p239 (starting at 320 µg/mL) and then incubated with ~64 copies per cell of HEV virions. In the neutralizing activity assay, the virus was pre-incubated with 2-fold serially diluted mAbs or PBS at 37 °C for 60 min prior to incubation with HepG2 and PPH. The following procedures were the same for all three assays. After incubation at 37 °C for 60 min, the infected cells were washed three times with PBS and the samples were harvested. HEV RNA was purified and quantified by real-time RT PCR.

### Binding and penetration of VLP
HepG2, PPH, and PHH cells were pre-cultured on coverslips. Subsequently, p239 wildtype or p239 mutants were directly incubated with the cells at 37 °C for 1 h or 4 h. For blocking controls, p239 wild-type or p239 mutants were pre-incubated with mAb 8G12 (200 µg/mL) at 37 °C for 1 h prior to co-incubation with the cells. The binding and entry efficiency of VLPs was analyzed by immunofluorescence assay as described below.

### Immunofluorescence assay
For immunofluorescence assay, cells were washed three times with PBS, fixed with 4% (w/v) paraformaldehyde (Sigma Aldrich, MO, USA), and permeabilized with PBS containing 0.3% Triton X-100 for 15 min. Subsequently, samples were blocked with 5% BSA in PBS for 1 h at room temperature. For VLP binding and entry assays, cells were incubated with mAbs 15B2 and 8C11 for 30 min at room temperature. For HEV virus infection assays, cells were incubated with mAbs 15B2 and 8G12 for 60 min at room temperature. After washing with PBS, fluorescein isothiocyanate-conjugated goat anti-mouse (GAM) IgG (Molecular Probes, CA, USA) was added and incubated for 30 min. Following five washes with PBS, samples were stained with 0.5% 4′, 6-diamidino-2-phenylindole (DAPI) (Invitrogen, CA, USA), and fluorescence signals were analyzed with an Opera Phenix, or Operetta CLS High-Content Analysis system (PerkinElmer, MA, USA).

### Viral infection detected by ELISA
Viral infection assay based on ELISA were performed as previously described[35]. HepG2/C3A cells or PPH were seeded into 96-well plates at a density of $3 \times 10^4$ cells per well one day before infection. Cells were inoculated with HEV (1000 copies per cell) at 37 °C for 12 h. For neutralizing assays, HEV wild-type or mutants were pre-incubated with mAb A286 or mAb 6H8 (200 µg/mL) at 37 °C for 1 h prior to incubation with the cells. Cells were washed three times with PBS and refed with complete medium supplemented with 2% DMSO. After incubation at 37 °C for 3 days, cell supernatants were collected and tested for HEV antigen using a commercial ELISA kit (Wantai, Beijing, China).

### Virus infection detected by immunofluorescence assay
Cells were seeded on 96-well glass-bottom plates (PerkinElmer, Inc., MA, USA) one day prior to viral infection. Viruses were diluted in culture medium containing 4% polyethylene glycol 8000 (PEG8000) to achieve 1000 copies per cell and then incubated with cells at 37 °C for 6 h. For neutralization assays, viruses were pre-incubated with mAb A286 or mAb 6H8 (200 µg/mL) at 37 °C for 1 h before infection. Cells were washed three times with PBS to remove unbound viral particles and subsequently cultured in complete medium supplemented with 2% DMSO for 5 days to allow viral replication and protein expression. Viral infection efficiency was evaluated using IFA as described above.

### Crystallization and structural determinations
The Fab fragment of mAb 6H8 was digested by papain and purified using anion-exchange chromatography with DEAE-5PW (TOSOH, Tokyo, Japan). The E2s-4:6H8 Fab complex was incubated at 37 °C for 2 h in a 1:1.5 molar ratio and then purified by size exclusion chromatography with G5000PWxl (TOSOH, Tokyo, Japan) using a buffer composed of 20 mM HEPES (pH 7.0). The E2s-4:6H8 Fab complex was concentrated to 8.5 mg/mL and used in crystallization experiments. The crystallization condition for E2s-4:6H8 Fab was 0.1 M Tris-HCl buffer (pH 8.5) with 25% w/v polyethylene glycol (PEG) 3350. All E2s proteins were concentrated to 13 mg/mL in crystallization experiments. Both E2s-1 and E2s-4-N490G/M492V were crystallized with a reservoir solution containing 0.1 M Bis-Tris buffer (pH 5.5) and 2 M $(NH4)_2SO_4$. The crystals of E2s-1-G490N/V492M were grown from a reservoir solution comprising 0.1 M NaAc and 2 M $(NH4)_2SO_4$. The reservoir solution for the crystals of 6H8 Fab was 20% PEG1500. All crystals were grown using the vapor diffusion technique in hanging drops at 21 °C, and 20% 2,3-Butanediol or 30% glycerol supplemented with their respective reservoir solutions were used as the cryoprotectant.

The X-ray diffraction data were collected at 100 K at the Shanghai Synchrotron Radiation Facility (SSRF) beam line 17U and processed using HKL2000 software. Subsequently, the structures were solved by molecular replacement using PHASER[65]. The model building was carried out in COOT[66], refined using the programs CNS[67] and PHENIX[68], and analyzed with PROCHECK[69]. Statistics for the data collection and structure refinement are summarized in Table S1.

### Cryo-EM and three-dimensional (3D) reconstruction of HEV-4 VLP:6H8 Complex
The HEV-4-p495:6H8 immune-complex was prepared by mixing of HEV-4 p495 VLP with oversaturated 6H8 Fab, following the corresponding crystal structure of the E2s-4:6H8 complex, with molar ratios of 1:1.2 for E2s monomer and 6H8 Fab. The complex was then incubated at 37 °C for 2 h. A 3 µL sample aliquot was deposited onto a glow-discharged Quantifoil holey carbon grid (R2/2, 200 mesh, Quantifoil Micro Tools). After 6 s of blotting to remove excess sample, the grid was plunge-frozen into liquid ethane using a Vitrobot Mark IV (Thermo Fisher Scientific). Images were recorded on a Falcon II direct detector camera (Thermo Fisher Scientific) equipped on a Tecnai F30 microscope (Thermo Fisher Scientific) at 300 kV, with a nominal magnification of 93,000, corresponding to a pixel size of 1.109 Å, and with defocus settings determined to be between 1.0 and 3.0 µm. For each image, the total electron dose was ~25 e/Å², with an exposure time of 1 s. Data were automatically collected using Thermo Fisher EPU software. A total of 1,066 micrographs were obtained and the contrast transfer function parameters of each micrographs were determined using Gctf[70]. Micrographs with excessive drift or astigmatism were discarded before reconstruction. Particles were manually picked using the program Eman2[71]. The origin and orientation parameters for each of these particle images were estimated using model-based procedures, and an initial model was generated with the random model method[72]. After several rounds of reference-free 2D and 3D classifications using Relion[73,74], a total of 5,901 good particles were selected for final 3D refinement with Relion. The map resolution was determined using the gold standard criteria of 0.143 Fourier shell correlation (FSC) cut-off[73]. Local resolution variations were estimated using the program ResMap[75]. The crystal structure was fitted into the corresponding map using the "fit in map" tool in UCSF Chimera[76]. Figures were prepared with PyMol[77] and UCSF Chimera.

### Molecular dynamics simulation
The MD models were derived base on the crystal structures of the of E2s-1, E2s-4, E2s-1-G490N/V492M, and E2s-4-N490G/M492V available in this paper. Another four single-site substitution structures (E2s-1-G490N, E2s-1-V492M, E2s-4-N490G and E2s-4-M492V) was modeled in the mutation wizard of PyMOL[77] based on the above crystal structures. Each E2s model was subjected to a 1000-ns simulation using the GROMACS 2024.2 software package[78], with CHARMM36 force field[79]. The protein was solvated in a cubic box with a minimum distance of

1.0 nm between the protein surface and the box edge using the TIP3P water model. Sodium (Na⁺) and chloride (Cl⁻) ions were added to neutralize the system and to achieve a final ion concentration of 150 mM. The systems were minimized for 1000 steepest-descent steps until the maximum force on any atom was less than 1000 kJ/mol/nm. After energy minimization, the system was equilibrated in two phases: a 200 picosecond (ps) NVT (constant number of particles, volume, and temperature) simulation at 310 K using the Nose-Hoover thermostat, followed by a 1 ns NPT (constant number of particles, pressure, and temperature) simulation at 1 bar using the Berendsen barostat. Subsequent production MD was performed NPT conditions at 310 K and 1 bar using the Parrinello-Rahman barostat. Long-range electrostatic interactions were calculated using the Particle Mesh Ewald (PME) method with a 1.2 nm cutoff for Coulomb interactions and van der Waals interactions. The LINCS algorithm was employed to constrain all bond lengths, enabling the use of a 2 femtosecond (fs) time step.

Trajectory data were saved every 10 ps for subsequent analysis. Analysis of the simulation trajectories, including RMSD and RMSF, was performed using GROMACS package. Visual molecular dynamics (VMD)[80] and MDAnalysis[81] were used to visualize and analyze the trajectories.

## Multiple sequence alignment

All available HEV protein sequences were retrieved from GenBank as of June 2025. Redundant and incomplete sequences were removed prior to analysis. For subsequent analyses, the region corresponding to the E2s domain was extracted from each aligned sequence. Sequence logo plots were created using WebLogo (https://weblogo.berkeley.edu)[82]. The extracted E2s domain alignments were submitted to WebLogo to generate graphical representations indicating amino acid conservation and variability at each position.

## Animal study

Cynomolgus monkeys (*Macaca fascicularis*) and Bama miniature swine were infected with HEV, and serial serum and stool samples were collected as previously reported[36,37]. Briefly, serum samples from cynomolgus monkeys (1.5–3 years old) and Bama miniature swine (2 months old) were screened for anti-HEV antibodies (anti-HEV IgM and anti-HEV IgG for cynomolgus monkeys; anti-HEV Ab for Bama miniature swine) (Wantai, Beijing, China), ALT (Maker Biotechnology, Sichuan, China), and HEV RNA prior to inoculation. Stool samples were suspended in 10% w/v PBS and screened for HEV RNA. HEV RNA was purified from 50 μL of each sample and quantified using real-time RT PCR as described above. Cynomolgus monkeys and Bama miniature swine with normal ALT levels and undetectable anti-HEV antibodies and HEV RNA were included in this study. Then, monkeys or swine ($n = 2$ per group) were inoculated with $1 \times 10^7$ copies of wild-type or mutant HEV, respectively. Serum samples were tested for anti-HEV antibodies, HEV RNA, and ALT, and stool samples were tested for HEV RNA. The results of the anti-HEV antibodies assays are shown as a signal-to-cutoff ratio (S/CO). Animal experimental protocols were conducted in accordance with the guideline of the Xiamen University Institutional Committee for the Care and Use of Laboratory Animals.

## Statistical analysis

Statistical analysis was carried out using GraphPad Prism version 8.0.1 (GraphPad Software Inc, CA, USA) and Origin 2021. $IC_{50}$ and $EC_{50}$ were calculated using a nonlinear regression model using GraphPad Prism. Data are presented as mean ± SD in Fig. 1A, B, F–H, and Fig. 7E–G, and as mean values in Fig. 2A, G, H, Fig. 4A, F, G, and Fig. 7A–D. Statistical analysis was performed with unpaired t-test (Fig. 1A, B) or two-way ANOVA (Fig. 1F–H). $P$ values were calculated from the two-tailed test, and $p$ values of 0.05 or lower were considered statistically significant. $P$ values are indicated by an asterisk (*) in plots: ns: not significant; *$p < 0.05$, **$p < 0.01$; ***$p < 0.001$; ****$p < 0.0001$.

## Reporting summary

Further information on research design is available in the Nature Portfolio Reporting Summary linked to this article.

## Data availability

The crystal structures generated in this study have been deposited in the Protein Data Bank (PDB) database under accession codes of 9L9Y (E2s-4:6H8), 9LB2 (E2s-1), 9LB3 (E2s-4-N490G/M492V), 9LB4 (E2s-1-G490N/V492M), and 9L9Z (6H8 Fab). The cryo-EM density map of the HEV-4-p495:6H8 has been deposited in the Electron Microscopy Data Bank (EMDB) under accession code EMD-62869. The nucleotide sequence of D32 has been deposited in the GenBank database under accession code PP475173.1 (https://www.ncbi.nlm.nih.gov/nuccore/PP475173.1/). Source data are provided with this paper.

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

## Acknowledgements

This work was supported by the National Natural Science Foundation of China (Grant No. 32370160 to Z-Z.Z, 82171746 to Z-M.T, 82402603 to D.Y), National Key Research and Development Program of China (Grant No. 2024YFC2310602 to Z-Z.Z), the Fujian Provincial Natural Science Foundation of China (Grant No. 2023J05263 to G-P.W and 2022J02005 to Z.Z.Z), the Fundamental Research Funds for the Central Universities (Grant No. 20720220004 and 20720220006 to N-S.X.), and the Xiamen Guiding Project for Medical and Health Care (Grant No. 3502Z20254ZD1296 to G-P.W). We thank the members in beamline BL17U1 at the Shanghai Synchrotron Radiation Facility for their assistance in X-ray data collection.

## Author contributions

Z-M.T., S-W.L., Q-B.Z., N-S.X., and Z-Z.Z. conceived the study and designed the experiments. Z-M.T., G-P.W., C.L., D.Y., Z-H.Y., M-Y. L., S-L.W., Z-H.C., and J-F.L. performed the experiments. Z-M.T., C-Y.Y., G-P.W., C.L., D.Y., H.Y., M-J.F., Y-B.W., J.Z., Y.G., Y.H., S-W.L., Q-B.Z., and Z-Z.Z. analyzed the data. Z-M.T., G-P.W., S-W.L., Q-B. Z., and Z-Z.Z. wrote the paper. All authors read and approved the final version of the manuscript.

## Competing interests

The authors declare no competing interests.

## Additional information

¹State Key Laboratory of Vaccines for Infectious Diseases, Xiang An Biomedicine Laboratory, School of Public Health, Xiamen University, Xiamen, China. ²National Institute of Diagnostics and Vaccine Development in Infectious Diseases, Collaborative Innovation Center of Biologic Products, National Innovation Platform for Industry-Education Integration in Vaccine Research, Xiamen University, Xiamen, China. ³NMPA Key Laboratory for Research and Evaluation of Infectious Disease Diagnostic Technology, School of Public Health, Xiamen University, Xiamen, Fujian, China. ⁴Department of Central Laboratory, Fujian Key Clinical Specialty of Laboratory Medicine, Women and Children's Hospital, School of Medicine, Xiamen University, Xiamen, Fujian, China. ⁵Department of Clinical Laboratory, Xiang'an Hospital of Xiamen University, School of Medicine, Xiamen University, Xiamen, China. ⁶These authors contributed equally: Zi-Min Tang, Cheng-Yu Yang, Gui-Ping Wen, Chang Liu, Dong Ying. ✉e-mail: shaowei@xmu.edu.cn; abing0811@xmu.edu.cn; nsxia@xmu.edu.cn; zhengzizheng@xmu.edu.cn

