## [Transparent Peer Review file · Nature Communications]

The Crucial but Insufficient Role of E2s Domain's Residues 490 and 492 in Determining the Host Tropism of Hepatitis E Virus

Corresponding Author: Dr Zizheng Zheng

Version 0:

Reviewer comments:

Reviewer #1

(Remarks to the Author)

The manuscript titled "Molecular Determinants of Hepatitis E Virus Host Tropism" investigates the molecular mechanisms underlying the host specificity of Hepatitis E virus (HEV) by focusing on the viral capsid protein E2 and its interaction with host cells. The study identifies key residues in the E2 protein that influence host specificity and virus-cell attachment, using a combination of structure-based mutagenesis, molecular dynamics simulations, and virus-cell attachment assays.

Major Concerns:

- The study's findings on the role of residues N490 and M492 in HEV host tropism are not sufficiently novel. Previous studies have already highlighted the importance of the E2s domain in HEV host specificity, and the current work does not significantly advance this understanding.
- The use of primary porcine hepatocytes (PPH) as a model for HEV study is problematic due to the lack of standardization and potential variability in cell isolation and culture conditions. This raises concerns about the reproducibility and reliability of the results.
- The molecular dynamics simulations, while extensive, do not provide conclusive evidence for the proposed mechanisms. The simulations should be complemented with additional experimental validation to strengthen the claims.
- The inoculums used in Figure 1 have not been characterized, as genome copy numbers are used instead of infection units or FFU. This approach is rather unspecific for determining HEV infection events, especially given that the authors appear capable of directly visualizing HEV via immunofluorescence.
- The in vivo experiments using Bama miniature swine and cynomolgus monkeys do not convincingly demonstrate the role of residues N490 and M492 in HEV host tropism. The failure of the HEV_1_G490N/V492M mutant to infect swine, despite regaining binding affinity to porcine hepatocytes, suggests that other factors are involved in host specificity. This undermines the central hypothesis of the study.
- The manuscript overinterprets the significance of the identified residues in determining HEV host tropism. The data do not support the conclusion that N490 and M492 are the primary determinants of HEV-4's specific tropism for porcine hepatocytes. Additional factors and mechanisms are likely involved, which are not addressed in the study.
- The study does not provide sufficient mechanistic insights into how the identified residues influence HEV host specificity. The proposed role of the N490-Y532 hydrogen bond in stabilizing the epitope conformation is speculative and requires further experimental validation.
- The manuscript does not adequately discuss the limitations of the study, including the potential variability in primary cell cultures, the limitations of molecular dynamics simulations, and the need for additional in vivo validation. The manuscript would benefit from a more thorough review of the existing literature to better contextualize the findings.
- The title "Molecular Determinants of Hepatitis E Virus Host Tropism" is rather unspecific and overstated and should be more precise.
- For some figures, it is unclear how often an experiment was conducted. It is essential to report in every figure legend how many technical or biological replicates were included. The authors should clearly state this as it is important to evaluate the robustness of the results. Additionally, it is often unclear how many cells were imaged or quantified or if images were taken from random areas. Information on this is also important for scientific soundness.

Specific Comments:

- Line 45: A more widely used term for the HEV capsid protein is P domain rather than E2.
- Line 52 and later in the manuscript: Change HEV_1 and HEV_4 to HEV-1 and HEV-4.
- Spelling and grammar checking suggested.

Figures and Experimental Controls:

- Figure 1: It is recommended to use primary human hepatocytes (PHH) as a more appropriate control for comparison to primary porcine hepatocytes (PPH). In panels A and B, inclusion of HEV-3 would provide a more comprehensive comparison.
- Figure 1C, D: Incorporating negative controls (untreated samples) and additional controls such as serum or other blocking agents would enhance the interpretability of the results.
- Figure 1E-F: The absence of blocking controls in these panels is noted. Including such controls would strengthen the conclusions drawn from these experiments.
- Figure 2G: It is suggested to add an ELISA-based readout in addition to the PCR-based readout to provide a more robust analysis.
- Line 188: Correct the reference from Fig 2G to Fig 2H.
- Figure 2H, I: Presenting neutralization data on authentic virus across different HEV genotypes would be beneficial. An immunofluorescence assay (IFA)-based readout, accompanied by respective IF images, would provide visual confirmation of the results.
- Supplementary Figures: Move Fig S5 to the main figures to ensure critical data is prominently displayed.
- Alanine Scanning: The study would benefit from alanine scanning of residues 488, 489, 490, and 532 on authentic virus and assess neutralization capacity. Additionally, investigate attachment not only in HepG2 cells but also in PHH to provide a more comprehensive analysis.
- Authentic Virus Data: Data from authentic virus experiments, currently in supplements S9E and F, should be moved to the main figures and discussed earlier in the manuscript to highlight their importance.
- Figure 6A: The inclusion of neutralization controls using antibodies or sera is recommended to validate the findings.
- Immunofluorescence Images: Ensure scale bars and magnification information are provided for all IF images to enhance clarity and reproducibility.
- Figure 6A: The use of authentic virus, rather than VLPs, would provide more relevant insights into viral behavior.
- Figure 6D, E: Include controls and present infection data, including IF images, using authentic virus to extend the study beyond mere binding analysis.
- Figure 6F: The authors should simplify the visualization of the data, enhance the x-axis legends, increase the number of animals in the study, and measure RNA levels in the liver to provide a more detailed and informative analysis.

Reviewer #2

(Remarks to the Author)

HEV has multiple genotypes: HEV1 and HEV2 infect only humans, while HEV3 and HEV4 infect a wide range of animal species including pigs other than humans. The basis of this difference in HEV's host range is not well understood. The study by Tang et al. identifies two amino acid residues (N490 and M492) on the HEV4 capsid critical for binding to an HEV3- and HEV4-specific monoclonal antibody 6H8 and infecting pig hepatocytes. Mutations in these residues in HEV4 abolished its ability to bind 6H8 and infect pig cells but had no effect on its ability to infect human cells. Conversely, substitution of these residues for those in HEV1 rendered HEV1 capable of infecting pig cells. However, HEV1 bearing N490 and M492 failed to establish infection in minipigs, indicating additional viral factors are also important for infection in pigs. This is an important study that advances our understanding of the viral determinants of HEV's host range. The manuscript is well-written. The data are of high quality and compelling. There are some minor issues that are listed below and should be addressed.

- Line 157 and Fig. 1G: Fig. 1G can be removed since no virions were detected in the cells.
- Fig. 3E: The amino acid alignment includes only one strain in each HEV genotypes. It would be good to include more HEV sequences to show the conservation of these residues.
- Line 263-264: Serine (S) should be Thr (T), and tyrosine (Y) should be Ser (S).
- Line 277: This sentence doesn't read well. How about "The structure...displayed greater similarity in the 480- and 530-loop to E2s_4 than E2s_1"?
- Fig. 5A: Change RMSF to RMSD on the y-axis.
- Line 358: Change formed to forming.
- Line 382: Change the cell lysate to the lysate of cells.
- Line 399: Change HEV_4 to HEV_1.

Reviewer #3

(Remarks to the Author)

This study provides insights into the molecular determinants of Hepatitis E virus (HEV) host tropism, particularly focusing on the zoonotic group HEV. The authors systematically identify key residues (N490 and M492) within the E2 domain of the capsid protein that contribute to HEV's specific binding to porcine hepatocytes. Various methodologies, including structure-based mutagenesis, molecular dynamics simulations, and virus-cell attachment assays, strengthen the conclusions drawn from their findings. Identifying the 6H8 antibody epitope, its structural characterization through crystallography and cryo-EM, and the exploration of mutant viruses in both in vitro and in vivo models are comprehensive.

However, there are areas for improvement.

1. Figure 3D: The labeling of amino acid residue appears quite crowded. The stick model, label text, and unnecessary shadows make it difficult for the reader to clearly distinguish the positions of key amino acids. A clearer layout or reducing label overlap would greatly improve the readability.
2. Figure 4: Several interactions lack distance annotations, which diminishes the clarity of the structural relationships between the residues.
3. Figure 5D: The use of PDB file Chain IDs as identifiers is understandable, but the two symmetric protomers of the E2s dimer do not appear to behave consistently, especially for E2s_1 V492M and E2s_4 N490G. This inconsistency is intriguing—does it suggest structural or functional differences between these protomers that could be relevant to the virus's mechanism or stability? It would be beneficial to provide a discussion of these discrepancies.
4. Figure 6E: There appears to be an absence of error bars.
7. Crystal Structure Model: The crystal structures, particularly E2s_4_N490G/M492V and E2s_1, may require further refinement. Notably, the Ramachandran plots for these two structures show a higher percentage of disallowed regions compared to the others (0.7% vs. 0%), which is unexpected. This could indicate either some special structural features arising from the crystallization conditions or simply an incomplete refinement process. It would be helpful for the authors to clarify whether these discrepancies stem from the protein crystallization energy state (e.g. packing interactions), or if further refinement was needed. Additionally, the authors' provided crystal structure PDB validation report is not as curated (NOT for manuscript review) as the cryo-EM structure validation (which was ready for manuscript review).

The results in this manuscript enhance the understanding of HEV's cross-species transmission potential, particularly regarding its zoonotic transmission. If the above deficiencies are addressed or clearly explained, I recommend this manuscript to be published in Nature Communications

Version 1:

Reviewer comments:

Reviewer #1

(Remarks to the Author)

The authors addressed many of the issues raised by the reviewers. Changing the title to "The Residues 490 and 492 on E2s Domain of the Capsid Protein Play a Pivotal but Insufficient Role in Determining the Host Tropism of Hepatitis E Virus" now describes the findings more accurately, nevertheless, is somewhat long and not elegant. Think about rephrasing.

It was very difficult to assess changes made to the manuscript, as new parts were only highlighted in yellow instead of tracking all changes including deletions and so on. Keep in mind for new submissions.

My major concerns still open are concerning the FFU measurements of the authors. Did you calculate the focus forming units per mL based on titrating the virus stocks? MOIs should reflect the number of infectious/intact particles per cell and mL. From literature we know that the ratio of RNA copies to infectious particles is roughly 1:1,000 (doi:

10.1073/pnas.1912307117). MOIs of 64 (Fig 1) or even 1,000 (p29 of revised manuscript) are extremely high and never have been reported for HEV before. Please make sure not to use RNA copies, but rather titrate the virus and assess FFU. Please also provide images of complete wells of your titration plate, usually 96well plate (compare e.g. doi: 10.1073/pnas.2202653119).

The sequence logos now presented in Fig S11 are somewhat confusing. The label for the y-axes are missing. Usually the height of the entire stack of residues is the information measured in bits. What do the authors mean by "degree of conservation"? How can conservation be the same over all residues here? Please explain and adapt.

Reviewer #2

(Remarks to the Author)

The revised manuscript has satisfactorily addressed my concerns.

Reviewer #3

(Remarks to the Author)

The authors revised the manuscript and responded one by one to all of my suggestions and concerns on structure, which was persuasive, although I still had doubts about the reliability of the "random" MD results and the extent to which they could be accepted to support the conclusions of the manuscript. I prefer to leave other virology and functional mechanisms to the other two reviewers to make judgments.

Version 3:

Reviewer comments:

Reviewer #1

(Remarks to the Author)

Thank you for addressing my concerns and correcting FFU. I hope the authors feel that their manuscript improved by this substantial revision.

Congratulations!

**Response to Reviewer Comments on the manuscript NCOMMS-25-13653-T:**

We would like to sincerely thank the reviews for their insightful comments, which have
significantly improved our manuscript. To facilitate the navigation of this document,
we have copied the reviewer's comments verbatim in **blue** and typed out our responses
in **black**. Thanks to the valuable suggestions, the manuscript now offers more insightful
information for readers.

**Reviewer #1 (Remarks to the Author):**

**General comments:**

The manuscript titled “Molecular Determinants of Hepatitis E Virus Host Tropism”
investigates the molecular mechanisms underlying the host specificity of Hepatitis E
virus (HEV) by focusing on the viral capsid protein E2 and its interaction with host
cells. The study identifies key residues in the E2 protein that influence host specificity
and virus-cell attachment, using a combination of structure-based mutagenesis,
molecular dynamics simulations, and virus-cell attachment assays.

**Response:** We sincerely thank the reviewer for the thoughtful evaluation and kind
recognition of our work. We truly appreciate these constructive feedback, which greatly
helped improve the clarity and quality of our manuscript. We have addressed all the
reviewer's comments and revised the manuscript accordingly. All changes in the
manuscript are highlighted in yellow.

**Major Concerns:**

1. The study's findings on the role of residues N490 and M492 in HEV host tropism are
not sufficiently novel. Previous studies have already highlighted the importance of the
E2s domain in HEV host specificity, and the current work does not significantly
advance this understanding.

**Response:** We appreciate the reviewer's recognition of prior work on the E2s domain's
role in HEV host specificity. Indeed, previous studies have demonstrated that the E2s

domain plays an important role in host specificity and swapping the E2s domain
between HEV-1 (human-restricted) and HEV-3 (zoonotic) alters infectivity in porcine
kidney cells *in vitro* (He et al, 2008, *J Gen Virol*; Nguyen et al, 2014, *J Virol*). However,
this approach left two critical gaps:

(1) Whether alteration of the E2s domain could disrupt the host's tropism of zoonotic
HEV for swine has not yet been verified *in vivo*.

(2) The functional mechanism of HEV host specificity remained unresolved. Prior work
did not identify specific residues governing tropism, nor did it explain why E2s domain
swaps only partially modulated infectivity.

This study bridges these gaps with three novel advances:

(1) We confirmed the critical role of E2s domain in determining HEV host specificity
both *in vitro* and *in vivo*.

(2) We identified of tropism-determining residues: through systematic mutagenesis, we
pinpointed N490 and M492 as the specific residues within E2s that dictate HEV-4's
ability to infect swine. Mutating these sites abolished HEV-4 infectivity in both porcine
cells (*in vitro*) and Bama miniature swine (*in vivo*), while reciprocal mutations in HEV-
1 restored its ability to infect porcine cells *in vitro* (**Fig. 7C-J**). This residue-level
resolution provides a mechanistic leap beyond domain-swap approaches.

(3) We performed *in vivo* tropism reprogramming: we showed that HEV-1 with
G490N/V492M mutations could not establish infection in piglets despite gaining *in*
*vitro* infectivity (**Fig. 7C-J**). This disconnect reveals that additional viral factors (e.g.,
ORF1, codon usage, or adaptive evolution) host factors (e.g., immune evasion or entry
receptors) constrain cross-species transmission *in vivo*.

We believed that these findings transform the E2s domain from a "black box" into a
functionally mapped determinant of HEV host range. We have revised the manuscript
on Page 6, line 110-113, Page 20, Page 17, line 422-431, Page 18, line 432-434, Page
20, line 501-503, line 513-514, Page 21, line 516-529, Page 30, line 782-795, Page 31,
line 796-814, to emphasize our findings.

2. The use of primary porcine hepatocytes (PPH) as a model for HEV study is
problematic due to the lack of standardization and potential variability in cell isolation
and culture conditions. This raises concerns about the reproducibility and reliability of
the results.

**Response:** We thank the reviewer for raising this important methodological
consideration. Primary cells provide an widely used model for studying HEV interplay
with host, including host response and host tropism. Since pigs represent the natural
host of HEV, the isolation of primary porcine hepatocytes (PPH) have been reported in
many previous studies (Chen, et al, 2019, *Hepatology*; Angelini et al, 2021, *Gut*; Todt
et al, 2020, *PNAS*; Li et al, 2024, *Biomed Pharmacother*; Nyberg et al, 2005, *Liver*
*Transpl*). PPH is the physiologically optimal model for zoonotic HEV-host interaction
studies, which have been used as a model of HEV study in previous studies (Todt et al,
2020, *PNAS*; Schlienkamp et al, 2025, *Emerg Microbes Infect*).

We understand the reviewer's concerns regarding the standardization of PPH models.
We performed strict culture criteria such as stringently restricted passage number (\leq
Passage 1) to avoid dedifferentiation. Furthermore, our findings on PPH were robustly
cross-verified. Therefore, we believe that our core findings on the PPH models is
reasonable.

3. The molecular dynamics simulations, while extensive, do not provide conclusive
evidence for the proposed mechanisms. The simulations should be complemented with
additional experimental validation to strengthen the claims.

**Response:** We thank the reviewer for highlighting the importance of experimental
validation for computational findings. Our main conclusion from the MD simulations
is that the N490-Y532 hydrogen bond critically stabilizes the 6H8 epitope conformation;
disruption of this bond, as seen in the G490 mutants, increases local flexibility and
abrogates 6H8 binding.

Importantly, this proposed mechanism is supported by both experimental and
computational data in our study. Specifically, binding assays (Fig. 4F-G) showed that

G490-containing mutants (E2-1, E2-1-V492M, E2-4-N490G, E2-4-N490G/M492V)
completely lose 6H8 binding, while variants retaining N490 (E2-4, E2-1-G490N, E2-
1-G490N/V492M, E2-4-M492V) preserve antibody recognition. Moreover, crystal
structures (Fig. 4H-I) corroborated the structural basis of this observation: the critical
N490-Y532 hydrogen bond is present in E2s-1 but absent in E2s-4. Thus, our study
provides both computational and experimental evidence that consistently support the
mechanism we propose.

4. The inoculums used in Figure 1 have not been characterized, as genome copy
numbers are used instead of infection units or FFU. This approach is rather unspecific
for determining HEV infection events, especially given that the authors appear capable
of directly visualizing HEV via immunofluorescence.

**Response:** We appreciate the reviewer's valuable comment. Based on the reviewer's
suggestion, we performed additional experiment to detect the infectivity of HEV on
HepG2 and PPH using immunofluorescence assay (IFA). As expected, the results
showed that the infectivity of HEV-3 and HEV-4 were observed both on HepG2 and
PPH, while infectivity of HEV-1 were observed exclusively on HepG2 but not on PPH.
In the revised manuscript, we have added the IFA-based results about HEV infection
on HepG2 and PPH (**Fig. 1C, Fig. S3, and Fig.S4**) and provided corresponding
descriptions at page 7, lines 143-149, Page 30, line 782-794, and Page 31, line 805-814.
In addition, we have changed the inoculums to multiplicity of infection (MOI) in the
revised in **Fig. 1A and 1B**.

**Revised Fig. 1C** Infectivity of HEV-1 and HEV-4 in HepG2 and PPH detected by
 immunofluorescence assay. Green fluorescence indicates HEV ORF2 protein
 expression, and blue fluorescence indicates 4',6-diamidino-2-phenylindole (DAPI)
 nuclear staining. Scale bar, 25 μ m.

5. The *in vivo* experiments using Bama miniature swine and cynomolgus monkeys do
 not convincingly demonstrate the role of residues N490 and M492 in HEV host tropism.
 The failure of the HEV_1_G490N/V492M mutant to infect swine, despite regaining
 binding affinity to porcine hepatocytes, suggests that other factors are involved in host
 specificity. This undermines the central hypothesis of the study.

**Response:** We appreciate the reviewer's insightful critique, which underscores the
 complexity of HEV host tropism. This study demonstrated that the residues N490 and
 M492 play a crucial but insufficient role in determining host specificity. While
 additional factors may modulate cross-species transmission, our data unequivocally
 demonstrate that N490 and M492 serve as non-redundant molecular switches. We
 provided loss-of-function proof by mutating N490/M492 in HEV-4 (a swine-adapted
 strain), which abolished infectivity of HEV in both porcine hepatocytes (*in vitro*) and
 Bama miniature swine (*in vivo*). We also provided gain-of-function proof by
 introducing N490/M492 mutations into HEV-1 (human-restricted), which restored
 binding and infectivity in porcine hepatocytes *in vitro*. This residue-level resolution

achieved through systematic mutagenesis across genotypes (HEV-1/4) provides the
direct evidence that N490/M492 are gatekeepers for swine entry.

The reviewer rightly notes that HEV-1-G490N/V492M failed to infect live swine
despite restored hepatocyte binding and infectivity. Our data show that N490/M492 are
necessary for the initial entry step, but post-entry factors (e.g., innate immune evasion
or replication machinery) may limit *in vivo* fitness. This aligns with known HEV
biology: many other viral factors such as ORF1, 5'UTR, codon usage, and adaptive
evolution, may be involved in determining HEV host tropism (Feagins et al, 2011,
*Virus Res*; Tian et al, 2020, *J Med Virol*; Chatterjee et al, 2016, *J Gen Virol*; Bouquet et
al, 2012, *Infect Genet Evol*; Nguyen et al, 2014, *J Virol*; Sun et al, 2020, *Infect Genet
Evol*; Lara et al, 2014, *Infect Genet Evol*; Primadharsini et al, 2019, *Viruses*;
Primadharsini et al, 2021, *Viruses*).

Considering that previous studies only correlated whole-domain swaps with tropism,
this study demonstrated that two-residue edits yield binary on/off phenotypes in cellular
models. To contextualize these mechanisms, we have revised the title to emphasize the
pivotal role of the two residues: “**The Residues 490 and 492 on E2s Domain of the
Capsid Protein Play a Pivotal but Insufficient Role in Determining the Host
Tropism of Hepatitis E Virus**”. We have also added more descriptions to discuss our
findings and limitations in the revised manuscript on Page 1, line 1-2, Page 5, line 93-
95, Page 6, line 96-98, Page 20, line 501-503, Page 21, line 517-529, Page 23, line 594-
599, and Page 24, line 600-602.

6. The manuscript overinterprets the significance of the identified residues in
determining HEV host tropism. The data do not support the conclusion that N490 and
M492 are the primary determinants of HEV-4's specific tropism for porcine hepatocytes.
Additional factors and mechanisms are likely involved, which are not addressed in the
study.

**Response:** We sincerely appreciate the reviewer's insightful critique regarding the
interpretation of our data. We acknowledge that the original manuscript contained

overstatements about the role of N490 and M492 residues in determining HEV-4 host
tropism. To address this concern, we have implemented the following modifications:

(1) We have revised the title to emphasize the pivotal role of the two residues more
rigorously: “**The Residues 490 and 492 on E2s Domain of the Capsid Protein Play**
**a Pivotal but Insufficient Role in Determining the Host Tropism of Hepatitis E**
**Virus**”.

(2) Content Refinements: N490 and M492 play a pivotal but insufficient role in
determining HEV host tropism, as mutation of these two residues disrupts both the
binding and infection of primary porcine hepatocytes in vitro and in vivo infection in
Bama miniature swine. Other viral factors, including ORF1, 5’UTR, codon usage, and
adaptive evolution, and host determinants, including host cellular factors and host
immune status, were also involved in determining HEV-4’s specific tropism for porcine
hepatocytes. We have further elaborated on the relevant descriptions in the
“Introduction” section on Page 5, line 93-95, Page 6, line 96-98, and “Discussion”
section on Page 20, line 501-503, Page 21, line 516-529.

(3) We also revised and provided a balanced conclusion for our findings: While
N490/M492 are essential for porcine hepatocyte infection, their function is likely
modulated by uncharacterized viral and host co-factors.

We believed that these revisions ensure our claims align with the experimental scope
while acknowledging the systemic complexity of tropism—a nuance we regret was
initially overstated.

7. The study does not provide sufficient mechanistic insights into how the identified
residues influence HEV host specificity. The proposed role of the N490-Y532 hydrogen
bond in stabilizing the epitope conformation is speculative and requires further
experimental validation.

**Response:** We thank the reviewer for this insightful critique. As emphasized in our
prior responses, we believe that our study is the first to resolve residue-level
determinants of HEV host tropism through an integrated structural and functional

approach. Below we want to address the mechanistic insights with enhanced clarity:
In this study, we identified that a zoonotic group HEV-specific epitope recognized by
mAb 6H8 was associated with HEV-4's specific tropism for porcine hepatocytes. The
binding assays of mutants with mAb 6H8 (**Fig. 4F-G**) showed that the G490-containing
mutants completely lose 6H8 binding, while variants retaining N490 preserve antibody
recognition. We also determined high-resolution ($<2 \text{ \AA}$) crystal structures of E2s-1,
E2s-4-N490G/M492V, and E2s-1-G490N/V492M, alongside the previously reported
E2s-4 structure. Crystal structures (**Fig. 4H-I**) showed that the critical N490-Y532
hydrogen bond is present in E2s-1 but absent in E2s-4. Given that crystal structures
represent time- and space-averaged states and are limited in capturing dynamic
conformational changes, we further performed molecular dynamics (MD) simulations
to elucidate potential molecular mechanisms. Our MD simulations across various E2s-
1 and E2s-4 wildtype and mutants consistently demonstrated a strong correlation
between the presence of the N490-Y532 hydrogen bond and epitope stability (**Fig. 5C**).
Thus, our findings suggest that the formation of the N490-Y532 hydrogen bond is a key
determinant in stabilizing the 6H8-recognized epitope, thereby influencing host tropism.

8. The manuscript does not adequately discuss the limitations of the study, including
the potential variability in primary cell cultures, the limitations of molecular dynamics
simulations, and the need for additional in vivo validation. The manuscript would
benefit from a more thorough review of the existing literature to better contextualize
the findings.

**Response:** In accordance with the reviewer's recommendation, we have added the
related description regarding the limitations of this study in the "Discussion" section on
Page 23, line 594-599, Page 24, line 600-602. We also provided a more detailed
description of the factors that determined the HEV host tropism on Page 5, Line 93-95,
Page 6, line 96-98, Page 20, line 501-503, and Page 21, line 516-529.

9. The title "Molecular Determinants of Hepatitis E Virus Host Tropism" is rather

unspecific and overstated and should be more precise.

**Response:** As emphasized in our prior responses, we have modified the title to “**The**
**Residues 490 and 492 on E2s Domain of the Capsid Protein Play a Pivotal but**
**Insufficient Role in Determining the Host Tropism of Hepatitis E Virus**” in the
revised manuscript.

10. For some figures, it is unclear how often an experiment was conducted. It is
essential to report in every figure legend how many technical or biological replicates
were included. The authors should clearly state this as it is important to evaluate the
robustness of the results. Additionally, it is often unclear how many cells were imaged
or quantified or if images were taken from random areas. Information on this is also
important for scientific soundness.

**Response:** Thanks for this important comment. The information about the replication
have been added in the corresponding figure legends. In particular, about 2000 cells
were quantified in **Fig. 6B** and **Fig. 6C**. In the **Fig. 7G**, about 69 imaged area were
quantified.

**Specific Comments:**

11. Line 45: A more widely used term for the HEV capsid protein is P domain rather
than E2.

**Response:** Revised as suggestion.

12. Line 52 and later in the manuscript: Change HEV_1 and HEV_4 to HEV-1 and
HEV-4.

**Response:** Based on the reviewer's suggestion, we have revised HEV_1, HEV_3, and
HEV_4 to HEV-1, HEV-3, and HEV-4, respectively, in the revised manuscript and
figures. We have also revised the E2_1, E2_2, E2_3, and E2_4 to E2-1, E2-2, E2-3, and
E2-4, respectively. And we have also revised p239_1, p239_3, and p239_4 to p239-1,
p239-3, and p239-4, respectively.

**Spelling and grammar checking suggested.**

**Figures and Experimental Controls:**

13. Figure 1: It is recommended to use primary human hepatocytes (PHH) as a more
appropriate control for comparison to primary porcine hepatocytes (PPH). In panels A
and B, inclusion of HEV-3 would provide a more comprehensive comparison.

**Response:** Thanks for the reviewers' suggestions, we are also interested in the binding
potency of HEV on PHH. Therefore, we have analyzed the binding of HEV on PHH as
well as the binding and penetration of VLP on PHH and added the related results in the
revised **Fig. S2** and **Fig. S5**. We have also added the related description in the revised
manuscript on Page 7, line 136, line 142-143, Page 25, line 655, Page 26, line 656-670,
Page 29, line 763, and Page 30, line 776. Meanwhile, we have supplemented the results
of the binding of HEV-3 on PPH and HepG2 in the revised **Fig. S1** and added the related
description in the revised manuscript on Page 7, line 136-137, line 139-140, line 142,
line 145.

**Revised Fig. S2. Binding potency of genotypes 1, 3, and 4 HEV to primary human**
**hepatocytes (PHH).** HEV-1: genotype 1 HEV; HEV-3: genotype 3 HEV; HEV-4:
genotype 4 HEV. Binding of HEV to PHH was blocked by cross-genotype neutralizing
monoclonal antibody 8G12. Data represent two biological replicates.

**Revised Fig. S5. Binding (1h) and penetration (4h) of genotype 1 p239 (p239-1)**
 **and genotype 4 p239 (p239-4) to primary human hepatocytes (PHH).** Green
 fluorescence indicates positive for p239 proteins; blue fluorescence indicates positive
 for 4',6-diamidino-2-phenylindole (DAPI) nuclear staining. Cross-genotype
 neutralizing monoclonal antibody (mAb) 8G12 was used to block the binding and
 penetration of p239 proteins. 1h+8G12 and 4h+8G12 indicate the binding and
 penetration of p239 proteins in the presence of mAb 8G12, respectively. The binding
 and penetration of p239-1 and p239-4 were blocked by mAb 8G12. NC (negative
 control) represents untreated cells. Scale bar, 25 μ m.

**Revised Fig. S1. Binding ability of genotype 3 HEV (HEV-3) to HepG2 and**
 **primary porcine hepatocytes (PPH).** Magenta column represents HEV-3 binding
 capacity to HepG2 cells, and cyan column represents HEV-3 binding capacity to PPH.
 Data represent three biological replicates. LOQ: limit of quantitation; MOI: multiplicity
 of infection.

14. Figure 1C, D: Incorporating negative controls (untreated samples) and additional
controls such as serum or other blocking agents would enhance the interpretability of
the results.

**Response:** Based on the suggestion of the reviewer, the negative control (NC) that were
not treated with p239 proteins and a cross-genotype monoclonal antibody 8G12 were
used as the blocking agents in the immunofluorescence assay of p239 wildtype. As
expected, the binding and penetration of both p239-1 and p239-4 on HepG2 and
primary porcine hepatocytes could be blocked by a previously reported HEV cross-
genotype neutralizing antibody 8G12. We have added the related results as **Fig. S6** in
supplement material and revised the description on Page 8, line 157-158, and Page 30,
line 778-779.

**Revised Fig. S6. Binding (1h) and penetration (4h) of genotype 1 p239 (p239-1)**
**and genotype 4 p239 (p239-4) on HepG2 cells (A) and primary porcine hepatocytes**

**(PPH) (B)**. Green fluorescence indicates positive for p239 protein; blue fluorescence
indicates positive for 4',6-diamidino-2-phenylindole (DAPI) nuclear staining. Cross-
genotype neutralizing monoclonal antibody (mAb) 8G12 was to block the binding and
penetration of p239 proteins on both cells. 1h+8G12 and 4h+8G12 indicate the binding
and penetration of p239 proteins in the presence of mAb 8G12, respectively. The
binding and penetration of p239-1 and p239-4 on both cell types were blocked by mAb
8G12. NC (negative control) represents untreated cells. Scale bar, 25 μ m.

15. Figure 1E-F: The absence of blocking controls in these panels is noted. Including
such controls would strengthen the conclusions drawn from these experiments.

**Response:** We apologize for the neglect. Following the comment, an irrelevant protein
was used as blocking controls and the corresponding results were presented in **Fig. 1F-**
**H** in the revised manuscript.

16. Figure 2G: It is suggested to add an ELISA-based readout in addition to the PCR-
based readout to provide a more robust analysis.

**Response:** We thank the reviewer for this valuable comment. Based on the suggestion,
the capture abilities of mAb 6H8 were analyzed using ELISA-based immune capture
assay. The results of ELISA-based immune capture assay demonstrated that mAb 6H8
presented capture ability for both HEV-3 and HEV-4, but not for HEV-1. The
corresponding results were added in **Fig. 2H**. We also revised the related description on
Page 9, line 198-199, and Page 29, line 748-752 in the revised manuscript.

H

**Revised Fig. 2H** Detection of different HEV genotypes captured by mAb 6H8 in an

immune capture experiment with ELISA for HEV antigen quantitation. The dash line
indicates the cut-off value. Data represent two biological replicates.

17. Line 188: Correct the reference from Fig 2G to Fig 2H.

**Response:** As the result of ELISA-based immune capture assay of mAb 6H8 have been
added in **Fig. 2H**, we have changed to **Fig. 2I** in the revised manuscript.

18. Figure 2H, I: Presenting neutralization data on authentic virus across different HEV
genotypes would be beneficial. An immunofluorescence assay (IFA)-based readout,
accompanied by respective IF images, would provide visual confirmation of the results.

**Response:** In the original manuscript, the neutralization ability of mAb 6H8 against
different genotypes HEV were assessed only in HepG2. According to the reviewer's
suggestion, we further performed the neutralization assay of mAb 6H8 against HEV-3
and HEV-4 in PPH. The results showed that 6H8 showed similar neutralizing activity
against HEV-3 and HEV-4 on PPH compared with those on HepG2. We have added the
corresponding results in the revised **Fig. 2J** and **Fig. 2L** (see attached below), and
revised the description at Page 9, line 204-206, page 10, line 208-209.

As of the IFA-based neutralizing assay, a robust cell culture model for different
genotypes HEV infection were currently absent (Todt et al, 2020, *PNAS*; Brüggemann
et al, 2025, *Trends Microbiol*). Most HEV study based on immunofluorescence assay
were rely on a specific HEV-3 strain (Kernow), which were derived from an
immunosuppressed patient with chronic HEV infection (Todt et al, 2020, *PNAS*; Liu et
al, 2019, *Viruses*; Yin et al, 2016, *J Virol*; Schlienkamp et al, 2025, *Emerg Microbes*
*Infect*). According to these studies, we could analyze the neutralizing activity of mAb
6H8 against HEV-1, HEV-3, and HEV-4 using a single-concentration IFA-based assay,
with mAb A286 serving as the reference control. The results demonstrated that mAb
6H8 exhibited potent neutralizing activity exclusively against HEV-3, and HEV-4 on
HepG2 and PPH, but not against HEV-1 on HepG2. We have added these results in the
revised manuscript at Page 10, line 209-211, and Page 31, line 805-814, and **Fig. S3**

and Fig. S4 (see attached below).

**Revised Fig. 2J and 2L (J)** Neutralizing activity of mAb 6H8 for different genotypes
of HEV in primary porcine hepatocytes (PPH). Data represent two biological replicates.

**(K)** Neutralizing activity of mAb 8G12 for different genotypes of HEV on HepG2. Data
represent two biological replicates. **(L)** Neutralizing activity of mAb 8G12 against

different genotypes of HEV in PPH. Data represent two biological replicates. mAb
8G12 with cross-genotype neutralizing capability for HEV was used as a control for the

neutralizing assay.

**Revised Fig. S3. Infectivity of wild-type and mutant HEV viruses in HepG2 cells**
**assessed by immunofluorescence assay.** Green fluorescence indicates positive for

HEV ORF2 protein and blue fluorescence indicates positive for 4',6-diamidino-2-
phenylindole (DAPI) nuclear staining. +6H8 and +A286 represent HEV infectivity in
the presence of mAb 6H8 and mAb A286, respectively. All wild-type and mutant
viruses were neutralized by mAb A286. HEV-3, HEV-4, and HEV-1-G490N/V492M
were neutralized by mAb 6H8. Scale bar, 50 μ m.

**Revised Fig. S4. Infectivity of wild-type and mutant HEV viruses in primary**
**porcine hepatocytes (PPH) assessed by immunofluorescence assay.** Green
fluorescence indicates positive for HEV ORF2 protein and blue indicates positive for
4',6-diamidino-2-phenylindole (DAPI) nuclear staining. +6H8 and +A286 represent
HEV infectivity in the presence of mAbs 6H8 and A286, respectively. Only viruses
with demonstrable infectivity on PPH are shown. HEV-3, HEV-4, and HEV-1-
G490N/V492M were neutralized by both mAbs A286 and 6H8. Scale bar, 50 μ m.

19. Supplementary Figures: Move Fig S5 to the main figures to ensure critical data is
prominently displayed.

**Response:** As shown in **Fig. S5** in the original manuscript (**Fig. S12** in the revised
manuscript), the mutation at residues 488 and 532 completely abolished the binding of

p239-4 on PPH, as well as the binding of p239-1 and p239-4 on HepG2. These results
indicated that these two residues may play an important role in the receptor recognition
on HepG2 and PPH. In current, the genuine functional receptor for HEV remain
unknown (Wißing et al, 2021, *Trends Microbiol*; Wang et al, 2021, *Curr Opin Microbiol*;
Brüggenmann et al, 2025, *Trends Microbiol*). This study focused on molecular
mechanism determining the HEV host range and cross-species infection. Thus, we think
that the original **Fig. S5** (**Fig. S12** in the revised manuscript) is more suitable to be
placed in the supplement material.

**20. Alanine Scanning:** The study would benefit from alanine scanning of residues 488,
489, 490, and 532 on authentic virus and assess neutralization capacity. Additionally,
investigate attachment not only in HepG2 cells but also in PHH to provide a more
comprehensive analysis.

**Response:** As shown in **Fig. S12**, the mutations at residues at 488 and 532 could
completely abolished the binding of HEV VLP p239 to both primary porcine
hepatocytes (PPH) and HepG2, indicating that residues 488 and 532 may be implicated
in HEV receptor recognition. The primary objective of this study was to elucidate the
molecular mechanisms underlying HEV cross-species infection, the roles of residues
488, 489, 490, and 532 on both PPH and HepG2 were not comprehensively investigated
in this study, and alanine-scanning mutant viruses targeting residues 488, 489, 490, and
532 were not generated, and their neutralization capacity were not assay. Due to the
limit of primary human hepatocytes (PHH), we investigated the attachment of only
p239 wildtype but not p239 mutants on PHH.

**21. Authentic Virus Data:** Data from authentic virus experiments, currently in
supplements S9E and F, should be moved to the main figures and discussed earlier in
the manuscript to highlight their importance.

**Response:** According to the reviewers' suggestions, we have moved **Fig. S9E** and **Fig.**
**S9F** in the supplemental material of the original manuscript to **Fig. 7A** and **Fig.7B** in

the revised manuscript. We have revised the description on Page 17, line 407-410.

22. Figure 6A: The inclusion of neutralization controls using antibodies or sera is
recommended to validate the findings.

**Response:** According to the reviewer's suggestion, a cross-genotype mAb 8G12 was
used as the neutralization control to validate the findings. We have revised the
description on Page 30, line 778-779. We added the related results performed on PPH
and HepG2 in **Fig. S16** and **Fig. S17**, respectively.

23. Immunofluorescence Images: Ensure scale bars and magnification information are
provided for all IF images to enhance clarity and reproducibility.

**Response:** Thanks for the comment. According to the reviewer's suggestion, we
have provided scale bars in the figures with immunofluorescence images, including **Fig.**
**1C-1E, Fig. 6A, Fig. 7H, Fig. S3, Fig. S4, Fig. S6, Fig. S12, Fig. S16, and Fig. S17.**
We also provided the information about the scale bar in the corresponding figure
legends on Page 42, line 1169, 1172, 1174, Page 45, line 1268, Page 46, line 1293.

24. Figure 6A: The use of authentic virus, rather than VLPs, would provide more
relevant insights into viral behavior.

**Response:** We are also interested in this issue. An amount of 10,000 MOI of HEV-1,
HEV-4, HEV-1-G490N/V492M, and HEV-4-N490G/M492V were added to the HepG2
and primary porcine hepatocytes (PPH), respectively, and then incubated at 37°C for 4
436 hours. The cells were then analyzed through immunofluorescence assay and
437 fluorescence images were acquired with Leica SP8 (Leica, Wetzlar, Germany). The
438 results showed that faint fluorescence signals were observed (see **Appendix Fig. 1**
attached below). Due to the lack of efficient HEV cell culture systems for different
genotypes, it is difficult for us to provide relevant insights for all wildtype and mutant
viruses at both 1h-incubation and 4h-incubation.

Appendix Fig. 1. The penetration of HEV in HepG2 and primary porcine

hepatocytes (PPH) detected by immunofluorescence assay. Scale bar: 50 μ m. NC

(negative control) represents untreated cells.

25. Figure 6D, E: Include controls and present infection data, including IF images, using

authentic virus to extend the study beyond mere binding analysis.

Response: We thank the reviewer for their valuable suggestion. As recommended, we

have assessed the infectivity of mutant and wild-type viruses in PPH and HepG2 cells

using ELISA and immunofluorescence assay, with mAb A286 serving as a neutralizing

control. The results showed that HEV-1, HEV-4-N490G/M492V, HEV-4 and HEV-1-

G490N/V492M exhibited similar infectivity on HepG2, while HEV-4 and HEV-1-

G490N/V492M have similar infectivity on PPH and HEV-1 and HEV-4-

N490G/M492V presented undetectable infectivity on PPH. These data have been added

in the revised manuscript on page (**Fig.7E-7H, Fig. S3 and Fig. S4**). We also added the

description on Page 16, line 385, Page 17, line 422-431, Page 28, line 432-434, Page

30, line 782-795, and Page 31, line 796-814.

 **Revised Fig. 7E-7H (E) Infectivity of wild-type and mutant viruses in PPH and**
 **HepG2 cells detected by ELISA for HEV ORF2 level.** Data represent three biological
 replicates. **(F) Ratio of HEV levels in the supernatant of PPH and HepG2 cells after**
 **HEV infection.** **(G) Number of positive cells detected by immunofluorescence**
 **assay after HEV infection.** Approximately 69 imaging areas were quantified. Data
 represent three biological replicates **(H) Infection capability of mutant viruses in**
 **PPH and HepG2 cells detected by immunofluorescence assay.** Green fluorescence
 indicates HEV ORF2 protein expression and blue fluorescence indicates positive for
 4',6-diamidino-2-phenylindole (DAPI) nuclear staining. Scale bar, 25 μ m.

26. Figure 6F: The authors should simplify the visualization of the data, enhance the x-
 axis legends, increase the number of animals in the study, and measure RNA levels in
 the liver to provide a more detailed and informative analysis.

**Response:** The animal experiments were conducted as previously described (Feagins
 et al, 2011, *Virus Res*; Gu et al, 2015, *Cell Res*; Tang et al, 2015, *Sci Rep*). During the
 animal experiments, the animals were monitored for viral shedding and the liver tissues

from these animals were not collected. Thus, we were unable to measure RNA levels
in the liver. As recommended by the reviewer, we have simplified the visualization of
the data concerning animal experiments (please referred to the revised **Fig. 7I** and **Fig.**
**7J**) and revised the corresponding figure legends on Page 46, line 1296, and Page 47,
line 1297, line 1299-1300.

**Reviewer #2 (Remarks to the Author):**

**General comments:**

HEV has multiple genotypes: HEV1 and HEV2 infect only humans, while HEV3 and
HEV4 infect a wide range of animal species including pigs other than humans. The
basis of this difference in HEV's host range is not well understood. The study by Tang
et al. identifies two amino acid residues (N490 and M492) on the HEV4 capsid critical
for binding to an HEV3- and HEV4-specific monoclonal antibody 6H8 and infecting
pig hepatocytes. Mutations in these residues in HEV4 abolished its ability to bind 6H8
and infect pig cells but had no effect on its ability to infect human cells. Conversely,
substitution of these residues for those in HEV1 rendered HEV1 capable of infecting
pig cells. However, HEV1 bearing N490 and M492 failed to establish infection in
minipigs, indicating additional viral factors are also important for infection in pigs.

This is an important study that advances our understanding of the viral determinants of
HEV's host range. The manuscript is well-written. The data are of high quality and
compelling. There are some minor issues that are listed below and should be addressed.

**Response:** We sincerely appreciate the reviewer for the positive comments. We have
addressed all the reviewer's comments and revised the manuscript accordingly. All
changes in the manuscript are highlighted in yellow.

1. Line 157 and Fig. 1G: Fig. 1G can be removed since no virions were detected in the
cells.

**Response:** Done as suggestion.

2. Fig. 3E: The amino acid alignment includes only one strain in each HEV genotypes.
It would be good to include more HEV sequences to show the conservation of these
residues.

**Response:** Good suggestion. We have now included most HEV sequences available in
GenBank to show the conservation of these highlighted residues. To maintain the clarity
and brevity of **Fig. 3**, we present the representative E2s sequences from HEV1-4 in the

main figure, and show the sequence logo plots, generated using WebLogo, in **Fig. S11**.
 The sequence logos graphically illustrate the conservation and variability of amino
 acids at each position. The methods for multiple sequence alignment and logo
 generation are described in detail in the Methods section (Page 34, line 890-897).

 **Revised Fig. S11. Sequence logo representation of the E2s region (residues 459-606)**
 **of HEV genotypes 1-4. (A)** Sequence logo representing the E2s region of genotype 1
 HEV (HEV-1) (n = 22) and genotype 2 HEV (HEV-2) (n = 1) strains. **(B)** Sequence
 logo representing the E2s region of HEV genotype 3 (n = 230) and genotype 4 (n = 6)

strains. The height of each letter at a given position represents the relative frequency of
the corresponding amino acid, and the overall height of the stack indicates the degree
of conservation at that position. Red arrowheads indicate critical amino acid residues
at positions 490 and 492.

**3. Line 263-264: Serine (S) should be Thr (T), and tyrosine (Y) should be Ser (S).**

**Response:** We are sorry for the negligence. We have changed Serine (S) to Thr (T) and
tyrosine (Y) to Ser (S), respectively, in the revised manuscript.

**4. Line 277: This sentence doesn't read well. How about "The structure...displayed
greater similarity in the 480- and 530-loop to E2s_4 than E2s_1"?**

**Response:** Thank you for your thoughtful feedback for this sentence. We agree that the
original sentence did not express our opinion well. Our original intention was to express
that the conformations of the 480- and 530-loop regions in E2s-1-G490N/V492M are
closer to those in E2s-4 than to E2s-1, which is reflected by lower RMSD values. We
agree that the sentence could be made clearer. Following your advice, we have revised
the sentence as follows to improve readability and explicitly highlight the comparison:
"The structural conformation of the 480- and 530-loop regions (6H8 epitope region) in
the structure of E2s-1-G490N/V492M is more similar to that in E2s-4 than to E2s-1, as
evaluated by root mean square deviations (RMSDs) of 0.227 and 0.458 Å." (Page 13,
lines 298-300)

**5. Fig. 5A: Change RMSF to RMSD on the y-axis.**

**Response:** Thank you for your careful reading and constructive feedback. We have
carefully reviewed Fig. 5A and would like to clarify that our intention in this figure was
to represent residue-specific flexibility. Therefore, we used root mean square
fluctuation (RMSF), which reflects the per-residue variations throughout the MD
simulation, rather than root mean square deviation (RMSD), which describes the
overall structural deviation. We believe RMSF is more appropriate for this purpose. To

prevent potential confusion, we added a brief introduction of the calculation of RMSF
in the revised manuscript for clarity: “To quantitatively compare the flexibility of
different structures, we calculated the root mean square fluctuation (RMSF) for each
residue based on the MD simulation trajectories.” (Page 14, line 325-327)

6. Line 358: Change formed to forming.

**Response:** Done as suggestion.

7. Line 382: Change the cell lysate to the lysate of cells.

**Response:** Done.

8. Line 399: Change HEV_4 to HEV_1.

**Response:** Done.

**Reviewer #3 (Remarks to the Author):**

This study provides insights into the molecular determinants of Hepatitis E virus (HEV)
host tropism, particularly focusing on the zoonotic group HEV. The authors
systematically identify key residues (N490 and M492) within the E2 domain of the
capsid protein that contribute to HEV's specific binding to porcine hepatocytes. Various
methodologies, including structure-based mutagenesis, molecular dynamics
simulations, and virus-cell attachment assays, strengthen the conclusions drawn from
their findings. Identifying the 6H8 antibody epitope, its structural characterization
through crystallography and cryo-EM, and the exploration of mutant viruses in both in
vitro and in vivo models are comprehensive.

**Response:** We sincerely appreciate the reviewer for the thoughtful evaluation and kind
recognition of our work. We truly appreciate these constructive feedbacks, which
greatly helped improve the clarity and quality of our manuscript. We have addressed all
the reviewer's comments and revised the manuscript accordingly. All changes in the
manuscript are highlighted in yellow.

However, there are areas for improvement.

1. Figure 3D: The labeling of amino acid residue appears quite crowded. The stick
model, label text, and unnecessary shadows make it difficult for the reader to clearly
distinguish the positions of key amino acids. A clearer layout or reducing label overlap
would greatly improve the readability.

**Response:** We have revised **Fig. 3D** to make the presentation clearer.

**Revised Fig.3D** Electrostatic potential surface of the epitope on E2s-4 (red, negative;
 blue, positive; and gray, neutral), with key residues for interaction from 6H8 CDR loops
 represented as sticks.

2. Figure 4: Several interactions lack distance annotations, which diminishes the clarity
 of the structural relationships between the residues.

**Response:** As suggestion, we have added the distance annotations in the revised **Fig.**

**4.**

3. Figure 5D: The use of PDB file Chain IDs as identifiers is understandable, but the
 two symmetric protomers of the E2s dimer do not appear to behave consistently,
 especially for E2s_1 V492M and E2s_4 N490G. This inconsistency is intriguing—does
 it suggest structural or functional differences between these protomers that could be
 relevant to the virus's mechanism or stability? It would be beneficial to provide a
 discussion of these discrepancies.

**Response:** We appreciate the reviewer's insightful observation regarding the apparent
 asymmetry between the two protomers of the E2s dimer in our MD simulations
 (original **Fig. 5D**), and thank you for raising this important point. While the asymmetry
 is indeed visually noticeable, we believe it does not reflect inherent structural or
 functional differences between the protomers. Rather, this variability arises from the
 stochastic nature of MD simulations. Although the two chains are structurally identical

in the starting model, minor differences in initial atomic velocities and local thermal
fluctuations can lead to asymmetric conformational behavior over the course of the
simulation, particularly in flexible regions such as the 6H8 epitope.
To evaluate whether this behavior was consistent, we performed two additional
independent simulation replicates for each variant under identical conditions. In all
cases, the degree of epitope deflection varied between the two chains. This supports the
conclusion that the observed asymmetry is a result of simulation randomness rather
than intrinsic protomer-specific properties.
Importantly, although the two chains may display differing degrees of deflection in a
given simulation, they exhibit the same overall trend across trajectories: variants
containing the N490G mutation consistently show greater epitope destabilization than
those retaining N490. We now include angular distribution frequency plots to better
illustrate this phenomenon in the revised **Fig. 5D**.

**Revised Fig. 5D** Distribution of deflection angles of the 480-loop region in different
 mutants in three replicate 1000-ns simulations. Each panel represents a distinct
 mutation, with six curves corresponding to three independent simulation runs (run1,
 run2, run3) for each chain (A and B) of the E2s dimer.

4. Figure 6E: There appears to be an absence of error bars.

**Response:** In the original manuscript, the average number of virions bound to PPH and
HepG2 was used to calculate the ratio and therefore without error bars. In the revised
manuscript, we change the data presentation by separately calculation the ratio of
virions bound to PPH and HepG2, enabling to display the error bars. We have update
the **Fig. 7C** with error bars in the revised manuscript.

5. Crystal Structure Model: The crystal structures, particularly E2s_4_N490G/M492V
and E2s_1, may require further refinement. Notably, the Ramachandran plots for these
two structures show a higher percentage of disallowed regions compared to the others
(0.7% vs. 0%), which is unexpected. This could indicate either some special structural
features arising from the crystallization conditions or simply an incomplete refinement
process. It would be helpful for the authors to clarify whether these discrepancies stem
from the protein crystallization energy state (e.g. packing interactions), or if further
refinement was needed. Additionally, the authors' provided crystal structure PDB
validation report is not as curated (NOT for manuscript review) as the cryo-EM
structure validation (which was ready for manuscript review).

**Response:** We sincerely appreciate the reviewer's thorough evaluation of our crystal
structures and the insightful questions raised regarding structural refinement. The two
structures mentioned by the reviewer (E2s-4-N490G/M492V at 0.95 Å, E2s-1 at 1.40
Å) do exhibit slightly higher Ramachandran disallowed percentages (0.7% vs. 0% in
other structures). However, this corresponds to only one residue per structure (see
**Appendix Fig. 2** and **3** attached below). Critically, we have Re-examined refinement
protocols and all steps (including TLS parameterization and iterative model rebuilding)
followed standard high-resolution refinement workflows. We have also validated
model-to-density fit and found that the disallowed residue showed unambiguous
electron density, confirming their accurate placement (see **Appendix Fig. 2** and **3**
attached below). We believed that, at such high (atomic) resolution, subtle
conformational strains—potentially induced by crystal packing interactions (e.g.,

intermolecular H-bonds)—may locally distort backbone geometry without indicating
refinement errors.

Appendix Fig. 2. The electron density map and fitted model representing to the disallowed residue of E2s-4-N490G/M492V in the Coot window

Appendix Fig. 3. The electron density map and fitted model representing to the disallowed residue of E2s-1 in the Coot window

Additionally, we apologize for the oversight in initially submitting non-curated
validation reports. Revised validation reports with "for manuscript review" status for
all crystal structures have now been uploaded to the submission system.

6. The results in this manuscript enhance the understanding of HEV's cross-species
transmission potential, particularly regarding its zoonotic transmission. If the above
deficiencies are addressed or clearly explained, I recommend this manuscript to be
published in Nature Communications

**Response:** We thank the reviewer for the careful reading of our manuscript, and
sincerely appreciate the encouraged comments for our study. According to the
reviewers' suggestions, we have included additional data to address all the reviewers'
comments in the revised manuscript.

Response to Reviewer Comments on the manuscript NCOMMS-25-13653A:

We thank the reviewers for the positive comments. We have addressed all the reviewer's comments and revised the manuscript accordingly. To facilitate the navigation of this document, we have copied the reviewer's comments verbatim in **blue** and typed out our responses in **black**.

Reviewer #1 (Remarks to the Author):

1. The authors addressed many of the issues raised by the reviewers. Changing the title to "The Residues 490 and 492 on E2s Domain of the Capsid Protein Play a Pivotal but Insufficient Role in Determining the Host Tropism of Hepatitis E Virus" now describes the findings more accurately, nevertheless, is somewhat long and not elegant. Think about rephrasing.

Response: We thank the reviewer for this insightful comment. In response to the reviewer's suggestion, we have revised the title to "**The Crucial but Insufficient Role of E2s Domain's Residues 490 and 492 in Determining the Host Tropism of Hepatitis E Virus**". We have revised the manuscript accordingly on page 1, lines 1-4.

2. It was very difficult to assess changes made to the manuscript, as new parts were only highlighted in yellow instead of tracking all changes including deletions and so on. Keep in mind for new submissions.

Response: We thank the reviewer for this comment. In response to this feedback, we have enabled Track Changes to ensure all revisions are clearly visible throughout the revised manuscript. Since track change mode of Word cannot highlight the modified content when generating a PDF, all changes were also highlighted in yellow.

3. My major concerns still open are concerning the FFU measurements of the authors. Did you calculate the focus forming units per mL based on titrating the virus stocks?

MOIs should reflect the number of infectious/intact particles per cell and mL. From literature we know that the ratio of RNA copies to infectious particles is roughly 1:1,000 (doi: 10.1073/pnas.1912307117). MOIs of 64 (Fig 1) or even 1,000 (p29 of revised manuscript) are extremely high and never have been reported for HEV before. Please make sure not to use RNA copies, but rather titrate the virus and assess FFU. Please also provide images of complete wells of your titration plate, usually 96well plate (compare e.g. doi: 10.1073/pnas.2202653119).

Response:

We thank the reviewer for these important questions regarding our viral quantification methodology. In this study, viral infectivity was assessed by enumerating positive cells through immunofluorescence assay (IFA) as previously reported (Liu et al, 2019, *Viruses*). We considered this enumeration method to be fundamentally analogous to the focus-forming unit (FFU) assay, as both approaches rely on immunofluorescence staining of infected cells to assess viral infectivity.

In this study, MOI refers to copies per cell rather than infectious particles per cell. For viral binding and infection assays, HEV virions were applied at 64 copies per cell or 1000 copies per cell. The use of copy numbers or genome equivalents (GE) as viral input measures has been widely employed in previous studies (Knegendorf et al, 2018, *Hepatol Commun*; Yin et al, 2016, *J Virol*; Yin et al, 2018, *PNAS*; Liu et al, 2019, *Viruses*). For instance, 25 GE per cell (Knegendorf et al, 2018, *Hepatol Commun*) or up to 10,000 GE per cell of HEV virions (Yin et al, 2016, *J Virol*; Yin et al, 2018, *PNAS*) were utilized. Furthermore, Liu et al. (2019, *Viruses*) demonstrated that 40-1000 GE per cell were required to achieve reliable IFA detection results. Notably, these studies were primarily based on the Kernow C1/p6 strain, which was isolated from a chronically HEV-infected patient (Shukla et al, 2011, *PNAS*). Many studies have demonstrated that the cell culture-adapted Kernow C1/p6 exhibits significantly higher infectivity and replication capacity compared to the primary isolate Sar55 (Shukla et al, 2011, *PNAS*; Knegendorf et al, 2018, *Hepatol Commun*). Given that the primary isolates of the genotype 1 and 4 strains (Sar55 and D32) used in this study inherently

demonstrate lower replication competence, the higher viral input is scientifically justified.

This study investigated the differences in binding and infectivity among different genotypes of HEV in human and porcine hepatocytes. The results demonstrated that, at equivalent copy-based viral input, the binding and infection efficiencies of different genotypes were comparable in human hepatocytes (**Fig. 1A-E, Fig. 7C-H, Fig. S2, and Fig. S3**). Similar levels of HEV antigen in supernatants and comparable numbers of IFA-positive cells were observed following HEV infection of the human hepatogenic cell line HepG2, indicating that the number of infectious particles was similar across different genotypes of HEV. We have revised the description accordingly in the revised manuscript on Page 29 line 770.

4. The sequence logos now presented in Fig S11 are somewhat confusing. The label for the y-axes are missing. Ususally the height of the entire stack of residues is the information measured in bits. What do the authors mean by "degree of conservation"? How can conservation be the same over all residues here? Please explain and addapt.

Response: We thank the reviewer for pointing this out and apologize for the ambiguity in our original presentation. In the original **Fig. S11**, the y-axis represented raw residue probabilities rather than information content (in bits), which may lead to confusion in interpreting sequence conservation. Following the reviewer's suggestion, we have revised **Fig. S11** to adopt the conventional representation, where the y-axis denotes information content in bits. Please refer to the updated **Fig. S11**, which is attached below.

Revised Fig. S11. Sequence logo representation of the E2s region (residues 459–606) of HEV genotypes 1–4. (A) Sequence logo representing the E2s region of HEV genotype 1 (n = 22) and genotype 2 (n = 1) strains. (B) Sequence logo representing the E2s region of HEV genotype 3 (n = 230) and genotype 4 (n = 6) strains. The height of each letter at a given position represents the information content (in bits) of the corresponding amino acid at that site. Red arrowheads indicate the critical amino acid residues at positions 490 and 492.

Reviewer #2 (Remarks to the Author):

The revised manuscript has satisfactorily addressed my concerns.

Response: We sincerely thank the reviewer for this positive comments.

Reviewer #3 (Remarks to the Author):

The authors revised the manuscript and responded one by one to all of my suggestions and concerns on structure, which was persuasive, although I still had doubts about the reliability of the “random” MD results and the extent to which they could be accepted to support the conclusions of the manuscript. I prefer to leave other virology and functional mechanisms to the other two reviewers to make judgments.

Response: We thank the reviewer for this follow-up feedback. We would like to reiterate that in this study, molecular dynamics (MD) was only used to assist us in our structural and biochemical findings, specifically regarding the role of the unique N490 in HEV-4 in stabilization of the epitope recognized by the zoonotic HEV-specific mAb 6H8. To address this, we performed three independent replicate MD simulations for each variant. Across all replicates, we observed highly consistent trends: variants containing G490 consistently exhibited increased epitope destabilization, while those containing N490 maintained structural stability. These results suggested that the observed effects are robust and not artifacts of random variation in the simulations. Therefore, we believe that the reproducibility and consistency of our findings across multiple independent runs provide strong support for the reliability of our conclusions. We acknowledge that stochastic fluctuations are inherent features of atomistic MD, reflecting the natural conformational variability of biomolecules. Nonetheless, we believe that our MD results corroborate our structural and biochemical findings, rather than drawing key conclusions solely based on MD analysis.